# Growth and adaptation mechanisms of tumour spheroids with time-dependent oxygen availability

**Ryan J. Murphy** [1]*, **Gency Gunasingh**[2], **Nikolas K. Haass**[2°], **Matthew J. Simpson**[1°]

1 Mathematical Sciences, Queensland University of Technology, Brisbane, Queensland, Australia, 2 Frazer Institute, The University of Queensland, Brisbane, Queensland, Australia

° These authors contributed equally to this work.

* r23.murphy@qut.edu.au

**Data Availability Statement:** Data Availability: The datasets generated and analysed during the current study are available on a GitHub repository (https://github.com/ryanmurphy42/Murphy2022SpheroidOxygenAdaptation) and are

## Abstract

Tumours are subject to external environmental variability. However, *in vitro* tumour spheroid experiments, used to understand cancer progression and develop cancer therapies, have been routinely performed for the past fifty years in constant external environments. Furthermore, spheroids are typically grown in ambient atmospheric oxygen (normoxia), whereas most *in vivo* tumours exist in hypoxic environments. Therefore, there are clear discrepancies between *in vitro* and *in vivo* conditions. We explore these discrepancies by combining tools from experimental biology, mathematical modelling, and statistical uncertainty quantification. Focusing on oxygen variability to develop our framework, we reveal key biological mechanisms governing tumour spheroid growth. Growing spheroids in time-dependent conditions, we identify and quantify novel biological adaptation mechanisms, including unexpected necrotic core removal, and transient reversal of the tumour spheroid growth phases.

## Author summary

Tumour spheroid experiments have been routinely performed for more than fifty years to understand cancer progression and develop cancer therapies. Spheroids are typically grown in fixed ambient atmospheric oxygen conditions whereas most *in vivo* tumours are subject to complicated fluctuating oxygen conditions. We explore this discrepancy between experimental conditions and *in vivo* conditions. In experiments we grow tumour spheroids subject to time-dependent oxygen conditions and, using new mathematical models and statistical uncertainty quantification, identify and quantify novel biological adaptation mechanisms. While tumour spheroid growth has traditionally been characterised by three phases of growth, we observe transient reversal of the tumour spheroid growth phases. These experimental observations are made possible by focusing on the time evolution of spheroids overall sizes and internal structure. To reveal the biological mechanisms underlying growth and adaptation in time-dependent oxygen conditions we extend the seminal Greenspan mathematical model. Unexpectedly, in time-dependent oxygen conditions we also observe necrotic core removal.

summarised in Text A in S1 File. Code availability: Key computer code and all experimental data used to generate computational results are available on a GitHub repository (https://github.com/ryanmurphy42/Murphy2022SpheroidOxygenAdaptation). The computer code for the mathematical modelling and statistical identifiability analysis was written in MATLAB R2021b (v9.11) with the Image Processing Toolbox (v11.4), Optimization Toolbox (v9.2), Global Optimization Toolbox (v4.6), and the Statistics and Machine Learning Toolbox (v12.2), and uses the *MCMCstat* package available on the GitHub repository (https://mjlaine.github.io/mcmcstat/).

**Funding:** MJS and NKH are supported by the Australian Research Council (DP200100177) https://www.arc.gov.au/. RJM is supported by the QUT Centre for Data Science https://research.qut.edu.au/qutcds/. The funders had no role in study design, data collection and analysis, decision to publish, or preparation of the manuscript.

## Introduction

*In vivo* tumours are subject to various types of environmental variability, for example due to fluctuating oxygen and nutrient availability [1–4]. To study cancer progression and develop cancer therapies, tumour spheroid experiments have been successfully and routinely performed for the past fifty years [2, 5–12]. However, tumour spheroid experiments are typically performed in constant environments and focus on the overall size of spheroids [2, 13–18]. By experimentally controlling oxygen availability and using mathematical modelling and statistical uncertainty quantification, we develop a new framework to study the impact of external environmental variability on the growth of tumour spheroids and their internal structure. Using our framework we identify and quantify novel biological adaptation mechanisms driven by environmental variability. This work begins to bridge the gap between *in vitro* and *in vivo* conditions, and lays the foundation for future experimental, mathematical, and statistical spheroid studies.

Oxygen availability is of particular importance since it is vital to the effectiveness of cancer therapies, such as chemotherapy and radiotherapy [1, 19, 20], and can be controlled in spheroid experiments. However, spheroid experiments are typically performed in ambient atmospheric conditions (21% oxygen), sometimes referred to as normoxia [13, 16]. In contrast, untreated tumours typically grow in variable hypoxic conditions (0.3–4.2% oxygen) [1, 20–22]. While many single-cell studies, and some spheroid studies, explore the role of environmental variability [2, 22–27], oxygen parameters critical to reproduce results are commonly not reported [28].

To visualise spheroid growth in normoxia, hypoxia, and time-dependent oxygen conditions we use fluorescent ubiquitination cell cycle indicator (FUCCI) transduced cell lines and hypoxia markers (Fig 1A–1E) [13, 14, 29, 30]: nuclei of cells in gap 1 (G1) phase fluoresce red, shown in magenta for clarity (Fig 1D); nuclei of cells in synthesis, gap 2, and mitotic (S/G2/M) phases fluoresce green (Fig 1D); and, regions of hypoxia are indicated by cyan (Fig 1B, 1C and 1E). Spheroids grown in constant normoxia experience three phases of growth (Fig 1A–1C and 1F) [10, 15, 16]. In phase (i) spheroids grow exponentially as all cells are able to proliferate, indicated by the presence of cells in the S/G2/M phases throughout the tumour spheroid shown by green (Fig 1A). In phase (ii) cells in the central region of the spheroid arrest in G1 phase while cells at the periphery continue to proliferate resulting in inhibited growth (Fig 1B). This arrested region is thought to arise due to spatial differences in nutrient availability, possibly oxygen, and/or a build up of metabolic waste from cells. In phase (iii) the spheroid is characterised by three regions: a central region composed of a necrotic core, $0 < r < R_n(t)$; an intermediate region of living but proliferation-inhibited cells, $R_n(t) < r < R_i(t)$; and, a region at the periphery composed of living and proliferating cells, $R_i(t) < r < R_o(t)$ (Fig 1C) [10, 13–16]. In comparison to spheroids grown in normoxia, spheroids grown in hypoxia form their necrotic core earlier, the distance from the edge of the spheroid to the hypoxic region and overall size are smaller (Fig 2).

To investigate environmental variability we perform additional experiments in time-dependent oxygen conditions. In these experiments we observe various tumour spheroid adaptation mechanisms (Fig 1H–1L). For instance, in re-oxygenation experiments we discover a novel adaptation process where the necrotic core of the spheroid that has formed prior to re-oxygenation moves within the spheroid and in certain situations exits the spheroid as a single object (Fig 1K and S1 Movie). Further, for fifty years tumour spheroid growth has been described by three sequential growth phases but re-oxygenation experiments show that spheroids can transiently experience these phases in the reverse order (Fig 1I and 1J). Other observations from

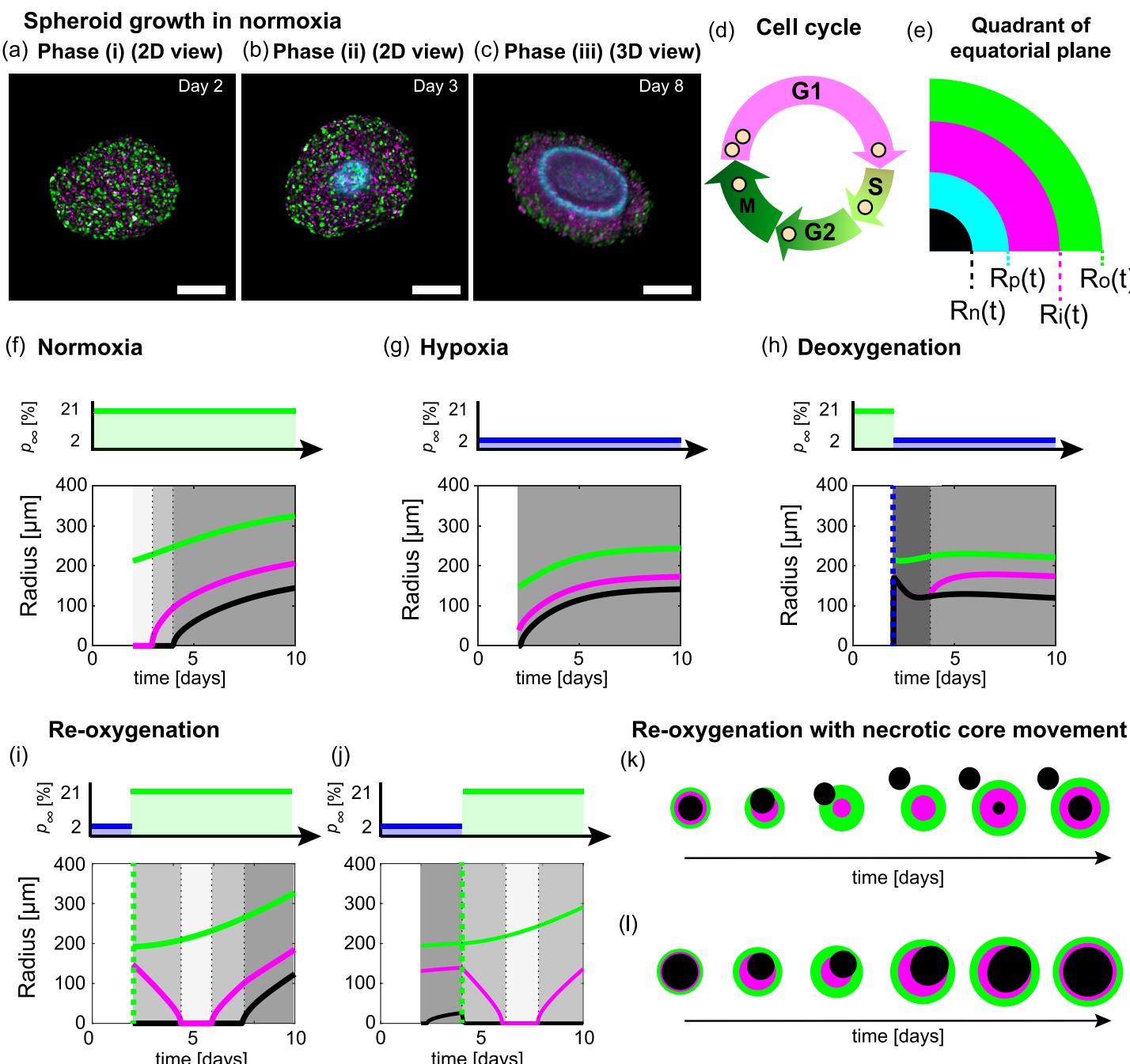

**Fig 1. Impact of external environment on the structure of growing tumour spheroids: A focus on oxygen availability.** (a-c) Tumour spheroid growth in standard experimental protocols occurs in three phases. Experimental images shown for FUCCI-transduced human melanoma WM983b spheroids grown in normoxia. (a-b) Experimental images of the equatorial plane of spheroids on Day 2 and 3 after seeding. (c) 3D z-stack representation of half of a spheroid on Day 8 after seeding. Scale bars are 200μm. Colours in (a-c) correspond to cell cycle schematic shown in (d): cells in G1 phase (magenta) and cells in S/G2/M phase (green). Pimonidazole staining reveals the hypoxic regions of spheroid (cyan). (e) Schematic for spherically symmetric spheroid structure representing a quadrant of the equatorial plane of a spheroid. Spheroids in normoxia experience three phases of growth, resulting in a spheroid with three regions at later times: a central region composed of a necrotic core, $0 < r < R_n(t)$ (black); an intermediate region of living but proliferation-inhibited cells, $R_n(t) < r < R_i(t)$ (magenta); and, a region at the periphery composed of living and proliferating cells, $R_i(t) < r < R_o(t)$ (green). The hypoxic radius, $R_p(t)$ (cyan) satisfies $R_n(t) \le R_p(t) \le R_o(t)$. (f-j) Schematics for oxygen conditions and time evolution of spheroid structure and overall size in (f) normoxia, (g) hypoxia, (h) deoxygenation experiments, and (i-j) re-oxygenation experiments where spherical symmetry is maintained. Note in (i-j) spheroids transiently undergo the growth phases in reverse. Greyscale shading in (f-j) represent growth phases. (k-l) Schematics for re-oxygenation experiments where we do not observe spherically symmetry due to (k) necrotic core removal and (l) movement of the necrotic core without removal. For additional experimental images for each condition with the boundaries of the proliferating, inhibited and hypoxic regions overlayed see Text A.2 in S1 File.

these experiments agree with intuitive expectations, but have not previously been explored nor quantified.

Many mathematical models describing tumour growth have been proposed and developed over the past fifty years, reviewed in [31–36]. Here we summarise studies that are of direct relevance to this study. First, we are interested in estimating the oxygen partial pressure profile within each individual spheroid at each time point. Grimes et al. [37] approach this problem using a mathematical model describing diffusion and consumption of oxygen. Using their oxygen diffusion model they estimate and validate the oxygen partial pressure within a spheroid, estimate the outer radius when the necrotic region forms, and the oxygen consumption rate. Gomes et al. [17] subsequently estimate the oxygen partial pressure within spheroids grown in normoxia (21% oxygen) and 5% oxygen which they refer to as physioxia. Their spheroid experiments, performed with HCT113 colon adenocarcinoma cells, show that spheroids grown in physioxia are smaller in overall size than spheroids grown in normoxia. Furthermore, Gomes et al. [17] measure the distance from the spheroid edge where 50% of the cells are proliferating and regions of hypoxia. They conclude that oxygen partial pressure is a rate-limiting parameter for cell proliferation. Here we briefly follow the approach of Grimes et al. [37] to analyse spheroid snapshots and then extend this analysis by exploring the temporal evolution of spheroid structure.

Throughout we quantitatively analyse the temporal evolution of the overall size and structure of tumour spheroids using mathematical modelling and statistical uncertainty quantification, including profile likelihood analysis and Bayesian inference approaches. We start with the seminal Greenspan mathematical model [10, 15, 16, 34]. Greenspan's model describes the three phases of growth and is relatively simple in comparison to other models [31–36]. This simplicity is a great advantage. Each mechanism and parameter in Greenspan's model has a biologically meaningful interpretation. Further, these parameters can be identified and estimated with the experimental data that we collect in this study [15, 16], which is not the case for other more complicated mathematical models [38–40]. In addition, here we show that we can extend Greenspan's model to analyse growth in time-dependent external oxygen partial pressures while retaining physical and biologically insightful interpretations of results. Recent work by Lewin et al. [41] has extended Greenspan's model in a comparable manner. However, in their theoretical study the external environmental conditions are constant throughout time and rapid changes in the oxygen partial pressure within the spheroid and to spheroid structure are driven by a rapid loss in living cell volume due to radiation induced cell death. With our new models we reveal biological adaptation mechanisms in de-oxygenation and re-oxygenation experiments and estimate key biological parameters.

In the following we first analyse spheroid experiments in normoxia and hypoxia. Such experiments demonstrate that Greenspan's model describes the experimental data remarkably well. Further, that oxygen mechanisms accurately describe the growth and formation of the necrotic core, whereas other mechanisms, possibly waste mechanisms, likely result in the growth and formation of the inhibited region. We then extend the mathematical model to interpret deoxygenation and re-oxygenation experiments, providing quantitative insights to biological adaption mechanisms throughout. We conclude by describing unexpected behaviours observed in re-oxygenation experiments.

## Results

Here we focus on WM983b spheroids in normoxia, hypoxia and deoxygenation experiments. Similar results for WM793b and WM164 spheroids are discussed in Text E and Text F in S1 File. For re-oxygenation experiments, we compare results from WM983b and WM793b

spheroids as we observe a range of behaviours. In Text F of S1 File we discuss additional WM164 re-oxygenation experiments.

## Oxygen diffusion alone is insufficient to describe spheroid growth

We capture end-point equatorial plane images for spheroids grown in normoxia and hypoxia measuring $R_o(t)$, $R_i(t)$, $R_n(t)$, and $R_p(t)$ (Fig 2A, 2C, 2D and 2F) (Methods: Image processing). These measurements are remarkably consistent within each condition and time point (Fig 2C and 2F). Comparing spheroids grown in normoxia and hypoxia, we observe vastly different tumour growth dynamics and internal structure ($\xi_n(t) = R_n(t)/R_o(t)$, $\xi_i(t) = R_i(t)/R_o(t)$, $\xi_p(t) = R_p(t)/R_o(t)$), even when comparing spheroids of similar size (Fig 2C and 2F, and Fig U in S1 File).

For deeper mechanistic insight we use mathematical modelling and statistical uncertainty quantification to interpret our observations. Specifically, we use Greenspan's mathematical model [10] and show that it accurately describes spheroid growth in both normoxia and hypoxia. Then using parameter estimation we identify biological mechanisms that differ between normoxia and hypoxia. Key model details are now discussed, for further details see Methods Greenspan's mathematical model and Text C.1 of S1 File. The model assumes each spheroid is spherically symmetric and maintained by cell-cell adhesion or surface tension. The independent variables are time $t$ [days], and radial position, $r$ [μm]. Conservation of volume gives an equation describing the time evolution of the outer radius, $R_o(t)$ [μm],

$$R_o^2(t)\frac{dR_o(t)}{dt} = \frac{s}{3}\left[R_o^3(t) - \max\left(R_i^3(t), R_n^3(t)\right)\right] - \lambda R_n^3(t), \tag{1}$$

where $s$ [day$^{-1}$] is the rate at which cell volume is produced by mitosis per unit volume of living cells, and $\lambda$ [day$^{-1}$] is the proportionality constant describing the rate at which cell volume is lost from the necrotic core. In these experiments, Eq (1) simplifies as $R_i(t) \geq R_n(t)$. This restricts our attention to two interpretations of the model that differ with respect to how $R_i(t)$ is defined. In the following discussion, we refer to these interpretations as *hypotheses* and show that hypothesis 2, where oxygen mechanisms drive $R_n(t)$ and waste mechanisms drive $R_i(t)$, is more consistent with the spheroids considered in this study.

In hypothesis 1 (Fig 3A), oxygen diffuses with diffusivity, $k$ [m$^2$ s$^{-1}$], and is consumed by living cells at a constant rate, $\alpha$ [m$^3$ kg$^{-1}$ s$^{-1}$]. The external oxygen partial pressure is $p_\infty$ [%]. Oxygen diffusion is fast relative to the growth of the spheroid, so that the oxygen partial pressure within the spheroid, $p(r(t))$ for $0 \leq r \leq R_o(t)$, is governed by

$$\frac{k}{r^2}\frac{\partial}{\partial r}\left(r^2\frac{\partial}{\partial r}p(r(t))\right) = \Omega\alpha H(r - R_n(t))\, H\left(R_o(t) - r\right), \quad 0 \leq r \leq R_o(t), \tag{2}$$

where $H(\cdot)$ is the Heaviside function and $\Omega$ [mmHg kg m$^{-3}$] is a conversion constant from volume of oxygen per unit tumour mass to oxygen partial pressure [37]. The inhibited radius, $R_i(t)$, is implicitly defined by $p(r(t)) = p_i$ [%] provided the oxygen partial pressure is sufficiently large (Fig 3A), and $R_i(t) = 0$ otherwise. Note we assume that $p_i$ [%] is a constant oxygen partial pressure independent of the external oxygen partial pressure.

In hypothesis 2 (Fig 3B), diffusible metabolic waste is produced by living cells at a constant rate per unit volume, $P$ [mol μm$^{-3}$ day$^{-1}$], and diffuses with diffusivity $\kappa$ [μm$^2$ day$^{-1}$]. Waste diffusion is fast relative to the growth of the spheroid, so the waste concentration within the

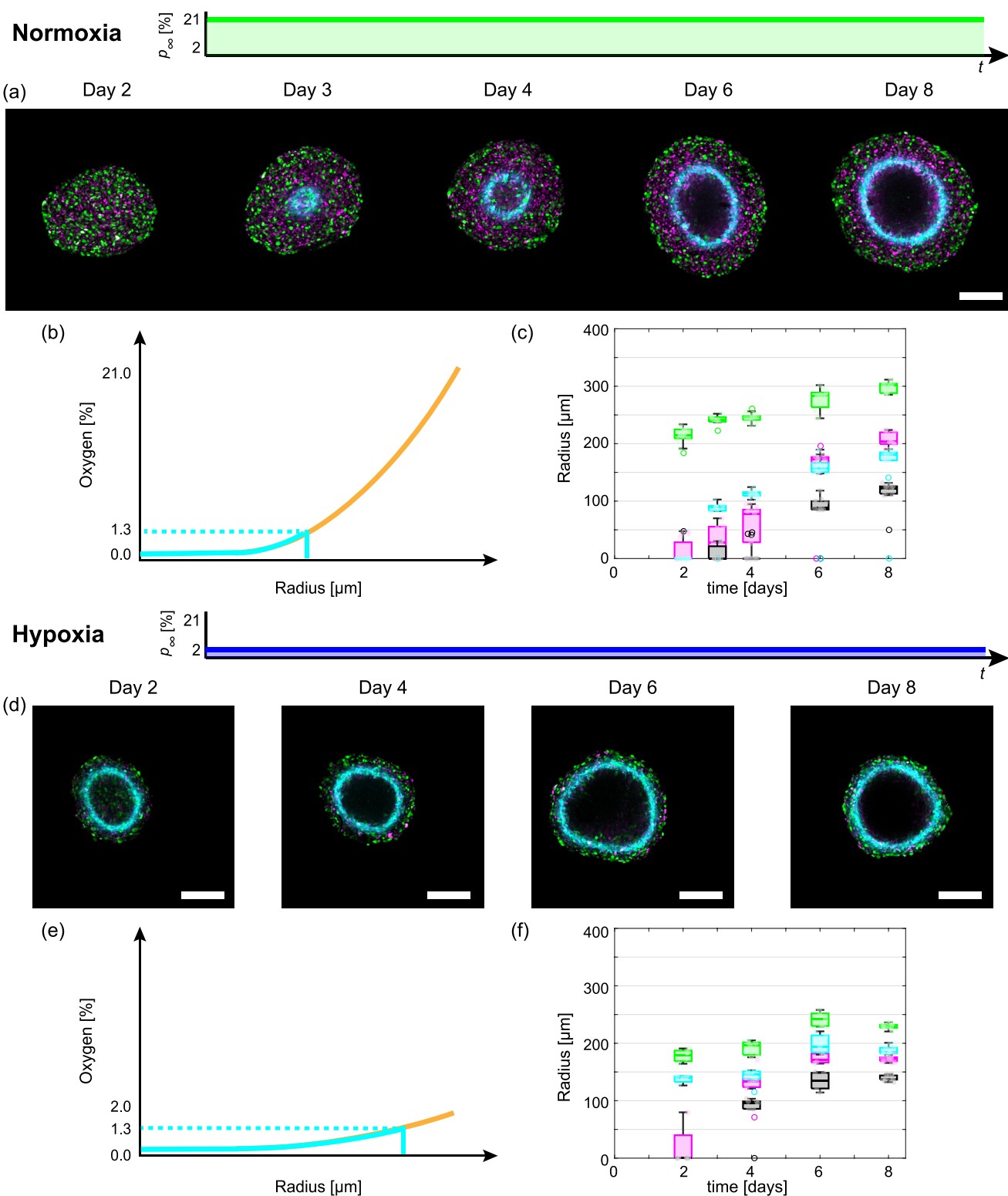

**Fig 2. Tumour spheroid growth in normoxia and hypoxia.** (a,d) Experimental images of the equatorial plane of FUCCI-transduced WM983b spheroids grown in (a) normoxia (21% oxygen) and (d) hypoxia (2% oxygen). (a) Images shown on Day 2, 3, 4, 6, and 8 after seeding. (d) Images shown on Day 2, 4, 6, and 8 after seeding. Scale bars are 200μm. (b,e) Schematics for oxygen partial pressure within spheroids for (b) normoxia and (e) hypoxia, where the solid blue/orange lines represent the oxygen partial pressure inside/outside of the hypoxic region and the hypoxic threshold is determined by the presence pimonidazole staining (blue-dashed). (c,f) Time evolution of $R_o(t)$ (green), $R_i(t)$ (magenta), $R_n(t)$ (black), and $R_p(t)$ (cyan) corresponding to schematic in Fig 1E for spheroids grown in (c) normoxia, and (f) hypoxia. Note that each spheroid measurement is an end-point measurement. For additional experimental images for each condition with the boundaries of the proliferating, inhibited and hypoxic regions overlayed see Text A.2 of S1 File.

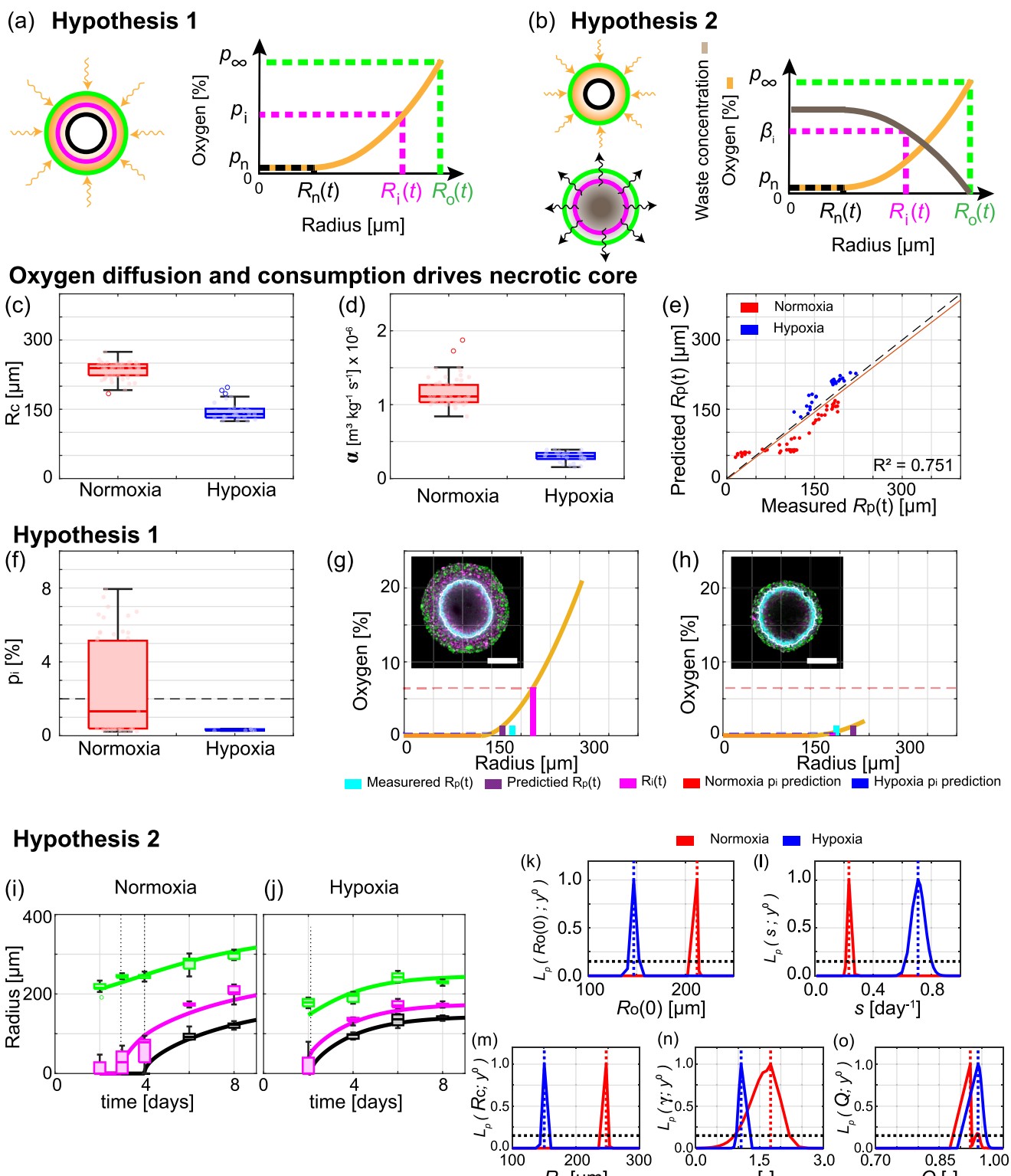

**Fig 3. Mechanisms governing tumour spheroid growth in normoxia and hypoxia.** (a,b) Schematics for two hypotheses. (a) Hypothesis 1: oxygen diffusion and consumption drives formation and growth of necrotic and inhibited regions. $R_n(t)$ [μm] and $R_i(t)$ [μm] implicitly defined via oxygen partial pressure thresholds $p_n$ [%] and $p_i$ [%], respectively. (b) Hypothesis 2: oxygen diffusion and consumption drives formation and growth of necrotic region, waste production and diffusion drives formation and growth of inhibited region. $R_n(t)$ [μm] implicitly defined via oxygen partial pressure thresholds $p_n$ [%]. $R_i(t)$ [μm] implicitly defined via waste threshold $\beta_i$ [mol μm$^{-3}$]. (c-h) Analysis of spheroid snapshots assuming hypothesis 1. Box charts representing

estimates of (c) outer radius when necrotic region forms, $R_c$ [μm], (d) oxygen consumption rate, $\alpha$ [m$^3$ kg$^{-1}$ s$^{-1}$], and (f) $p_i$ [%] for spheroids grown in normoxia and hypoxia. (e) Comparison of measured and predicted $R_p(t)$ [μm] ($R^2$ = 0.751) suggesting the estimated oxygen partial pressure within each spheroid is reasonable. (g,h) Estimated oxygen partial pressure within (g) a spheroid grown in normoxia and (h) a spheroid grown in hypoxia. Insets in (g, h) show FUCCI signal and pimonidazole staining from Day 8 and 6, respectively, with scale bar is 200 μm. In (g,h) horizontal red and blue dashed lines correspond to estimates of $p_i$ [%] from spheroid grown in normoxia and hypoxia, respectively. The estimate of $p_i$ from normoxia in (g) suggests the spheroid growing in hypoxia in (i) should be entirely arrested under hypothesis 1. (i-o) Analysis of temporal evolution of spheroid structure using Greenspans mathematical model and hypothesis 2. (i-j) Comparison of experimental data (box charts) with Greenspan's mathematical model (solid lines) simulated with the maximum likelihood estimates of parameters for (i) normoxia and (j) hypoxia. In (i,j) time evolution of outer radius, $R_o(t)$ (green), inhibited radius, $R_i(t)$ (magenta), and necrotic radius, $R_n(t)$ (black). Data represent an average of twelve spheroids on days 2, 3, 4, 6, and 8 for spheroids grown in normoxia and an average of seven spheroids on days 2, 4, 6, 8 for spheroids grown in hypoxia (Methods: Experimental methods). (k-o) Profile likelihoods for parameters of Greenspans mathematical model: (k) initial outer radius, $R_o$ [μm], (l) proliferation rate, $s$ [day$^{-1}$], (m) outer radius when necrotic region first forms, $R_c$ [μm], (n) dimensionless parameter relating proliferation rate and mass loss from necrotic core, $\gamma$ [-], and (o) dimensionless parameter relating oxygen and waste mechanisms, $Q$ [-] (Methods: Greenspan's mathematical model).

spheroid, $\beta(r(t))$ [mol μm$^{-3}$], is governed by

$$\frac{\kappa}{r^2}\frac{\partial}{\partial r}\left(r^2\frac{\partial}{\partial r}\beta(r(t))\right) = -P\mathrm{H}(r - R_n(t))\,\mathrm{H}\left(R_o(t) - r\right), \qquad 0 \le r \le R_o(t). \qquad (3)$$

The inhibited radius, $R_i(t)$, is implicitly defined through $\beta(r(t)) = \beta_i$ [mol μm$^{-3}$] provided the waste concentration is sufficiently large (Fig 3B), and $R_i(t) = 0$ otherwise.

Both hypothesis 1 and 2 assume that $R_n(t)$ is implicitly defined by $p(R_n(t)) = p_n$ provided the oxygen partial pressure is sufficiently small, and $R_n(t) = 0$ otherwise. Informed by experimental results [37], we set $p_n = 0$ [%].

Analysis of the model provides an analytical expression for the time when the inhibited region forms (Eq 7.1). For hypothesis 2, the inhibited region is predicted to form at the same time independent of oxygen dynamics, provided the spheroids grown in normoxia and hypoxia are initially the same size. This appears consistent with results in Fig 2A–2D where the inhibited region has formed on day 2 for both conditions. In contrast, with hypothesis 1 the time to form the inhibited region depends on oxygen mechanisms and specifically $p_\infty$, but without knowledge of additional parameters further insights are unclear.

By incorporating statistical uncertainty quantification to estimate parameters of the model we gain further mechanistic insights. A key assumption common to hypothesis 1 and 2 is that oxygen diffusion and consumption drives the time evolution of $R_n(t)$. We test this assumption directly by analysing radial measurements of each spheroid at each time point independently [37]. Using measurements of $R_o(t)$ and $R_n(t)$ we estimate: the outer radius when the necrotic region first forms, $R_c$; $\alpha$; and $R_p(t)$ (Fig 3C and 3D, and Text D.1.1 in S1 File). Image processing to measure $R_p(t)$ is more challenging than for $R_o(t)$ due to gradients in the pimonidazole signal (Text B in S1 File). However, we find good agreement between experimentally measured and predicted values of $R_p(t) > (R^2 = 0.751$, Fig 3E). This approach allows us to estimate the oxygen partial pressure within each spheroid at each time point (Fig 3G and 3H). From the results in Fig 3E, 3G and 3H, and Fig V in S1 File we conclude that oxygen of diffusion alone is a reasonable and sufficient mechanism to describe the formation of the necrotic core in WM983b spheroids [37]. Furthermore, these results suggest that the following assumptions are reasonable: $p_n = 0$; a constant oxygen consumption rate within the spheroid; spherical symmetry; oxygen at the edge of the spheroid can be approximated with the oxygen settings on the incubator.

Given that $R_n(t)$ is reasonably described by oxygen mechanisms we now examine hypothesis 1. Hypothesis 1 assumes that oxygen mechanisms alone drive the time evolution of $R_i(t)$. We then estimate $p_i$ using the estimated oxygen partial pressure within each spheroid, $p(r(t))$ for $0 < r < R_o(t)$, measurements of $R_i(t)$ and the definition $p(R_i(t)) = p_i$ (Fig 3A, 3G and 3H). Estimates of $p_i$ are consistently larger for spheroids grown in normoxia compared to spheroids

grown in hypoxia (Fig 3F), consistent with results for other cell lines [17]. This is inconsistent with hypothesis 1. Specifically, results from normoxia suggest that spheroids grown in hypoxia should have larger inhibited regions than experimentally measured (Fig 3G and 3H). Similarly, results from spheroids grown in hypoxia suggest that spheroids grown in normoxia should have smaller inhibited regions than experimentally measured. Further, we do not restrict ourselves to the notion that oxygen must be directly responsible for formation and growth of the inhibited region as it is unclear why cells within spheroids grown in normoxia arrest at a higher oxygen partial pressure than cells within spheroids grown in hypoxia. These results provide strong evidence to suggest that oxygen alone is insufficient to describe the formation of the inhibited region across multiple oxygen conditions for this cell line.

To test hypothesis 2, which assumes that waste mechanisms drive the time evolution of $R_i(t)$, we first analyse measurements of each spheroid at each time point independently. Experimentally measuring waste within spheroids is challenging, so we estimate the waste concentration within each spheroid and use measurements of $R_i(t)$ to estimate the outer radius when the inhibited region first forms, $\mathcal{R} = \beta_i \kappa / P$ (Fig W in S1 File). We observe that $\mathcal{R}$ is larger for spheroids grown in normoxia than hypoxia, which may be due to changes in $\beta_i$ or $P$ or $\kappa$ (Fig W in S1 File). These results do not provide sufficient evidence to reject hypothesis 2.

To test whether hypothesis 2 is reasonable we estimate model parameters for spheroids grown in normoxia and hypoxia. Specifically, we estimate the five key parameters: $\Theta_g = (R_o(0), s, R_c, \gamma, Q)$, where $\gamma = \lambda/s$ [-] and $Q$ [-] are dimensionless quantities (Methods: Parameter estimation and identifiability analysis). Simulating the model with the maximum likelihood estimate (MLE), $\hat{\Theta}_n$, for normoxia shows good agreement with the experimental data from spheroids grown in normoxia (Fig 3I). Similarly, simulating the model at $\hat{\Theta}_h$ for hypoxia shows good agreement with the experimental data from spheroids grown in hypoxia (Fig 3J). These results suggest the model accurately captures the dynamics of tumour spheroid growth.

Alongside the point estimates $\hat{\Theta}_n$ and $\hat{\Theta}_h$, we are interested in forming approximate 95% confidence intervals for each of the parameters. To perform this analysis we employ profile likelihood analysis (Methods: Parameter estimation and identifiability analysis). All profile likelihoods computed here are narrow and well-formed around a single central peak, the MLE, indicating that parameters are identifiable and that a relatively narrow range of parameters give a similar match to the data as the MLE (Fig 3K–3O) [16]. Approximate 95% confidence intervals are obtained from these profile likelihoods for each parameter. The profile likelihoods for the initial outer radius, $R_o(0)$, do not overlap and agree with observations that $R_o(0)$ is smaller for spheroids grown in hypoxia than normoxia (Fig 3K). The profile likelihood for $s$ interestingly estimates that the rate of cell proliferation per unit volume is faster in hypoxia than normoxia (Fig 3L). This result may seem surprising as a simplistic assumption would be that less oxygen results in less proliferation. However, our result is consistent with observations from other cell lines where an intermediate level of hypoxia encourages more proliferation than normoxia [42–44]. Profile likelihoods for $R_c$ are consistent with estimates of $R_c$ obtained by analysing spheroid measurements independently, a good consistency check for the two methods (Fig 3C and 3M). The other profile likelihoods for $\gamma$ and $Q$ overlap suggesting that these parameters are consistent across normoxic and hypoxic conditions (Fig 3N and 3O). Posterior densities and prediction intervals, estimated using Bayesian inference, also show good agreement with results here and the experimental data (Fig X in S1 File). Similar results hold for the other two cell lines (Text E and Text F in S1 File).

In the remainder of this study, we take the most fundamental approach and proceed with Greenspan's mathematical model and interpret the governing mechanisms with hypothesis 2. We note that other biological mechanisms may also be relevant. However, as the model already

appears to capture the key dynamics (Fig 3I and 3J) we will avoid overcomplicating the model at this stage. Furthermore, in the following analysis of deoxygenation and re-oxygenation experiments we necessarily extend the model.

## Adaptation to deoxygenation

The mechanisms underlying how tumour spheroids adapt to time-dependent external environments is unclear. Here, we perform a series of deoxygenation experiments, where spheroids grown in normoxia are transferred to hypoxic conditions at $t = t_s$ [days]. Analysing spheroid snapshots reveals how spheroids, and in particular their internal structure, adapt. Extending Greenspan's mathematical model and using parameter estimation, we identify and quantify key biological adaptation mechanisms.

In the deoxygenation experiments we set $p_\infty = 21$ [%] for $0 < t \le t_s$ [days], $p_\infty = 2$ [%] for $t_s < t < 8$ [days], and $t_s = 2$ [days] (Fig 4A). At $t_s = 2$ [days] all spheroids are in phase (i) of growth with proliferating cells throughout and no inhibited or necrotic region (Day 2 of Fig 4B). At $t_s + 1$ [days] the FUCCI signal in the central region of the spheroid is blurred, relative to the signal at the periphery, indicating dying and dead cells (Day 3 of Fig 4B and Fig T in S1 File). Therefore, we identify this central region as the necrotic core (Text B in S1 File). Experimental images at later times show that $R_n(t)$, $R_i(t)$ and $R_p(t)$ continue to increase but at a much slower rate (Days 4–8 of Fig 4B and 4C). Throughout the experiment $R_o(t)$ remains approximately constant (Days 2–8 of Fig 4B and 4C) confirming that the most important changes involve the internal structure and not the overall spheroid size. Further, $\xi_n(t)$, $\xi_i(t)$, and $\xi_p(t)$ approach values observed at late times for spheroids grown in hypoxia (Fig 4C and Fig Y in S1 File).

To interpret these deoxygenation experiments we extend Greenspan's mathematical model [10]. We assume that the change in $p_\infty$ at $t_s$ is instantaneous, which is reasonable since the switch from normoxia to hypoxia requires only 1–2 minutes when the spheroids are transferred between incubators. This time is very short in comparison to the duration of the experiment and time interval between data points. Similarly, we assume that the oxygen partial pressure within the spheroid adapts to the change in $p_\infty$ instantaneously, which is reasonable since oxygen takes approximately 10 seconds to diffuse across a distance of 100 μm [10]. Then we estimate the oxygen partial pressure within the spheroid at $t_s$ under normoxic and hypoxic conditions (Methods: Mathematical model to interpret deoxygenation experiments, Fig 4D). Immediately after $t_s$ the predicted necrotic radius, denoted $R_n^+(t)$ and implicitly defined by $p(R_n^+(t)) = 0$, is greater than the actual necrotic radius, $R_n(t)$, specifically $R_n^+(t) > R_n(t)$ (Fig 4D–4F).

Before considering the region $R_n(t) < r < R_n^+(t)$, recall that parameter estimates from spheroids grown in normoxia and hypoxia differ. Specifically, $\alpha$ (Fig 3D), $\lambda = \gamma s$ (Fig 3L and 3N), $s$ (Fig 3L), and $\mathcal{R}$ (Fig W in S1 File) are all different. Therefore, we expect that these parameter values will evolve in time after $t_s$. To account for such changes we define the following, for $t \ge t_s$,

$$\alpha(t) = \alpha_h + (\alpha_n - \alpha_h)\exp\left(-\frac{1}{\tau_\alpha}(t - t_s)\right), \tag{4.1}$$

$$\lambda(t) = \lambda_h + (\lambda_n - \lambda_h)\exp\left(-\frac{1}{\tau_\lambda}(t - t_s)\right), \tag{4.2}$$

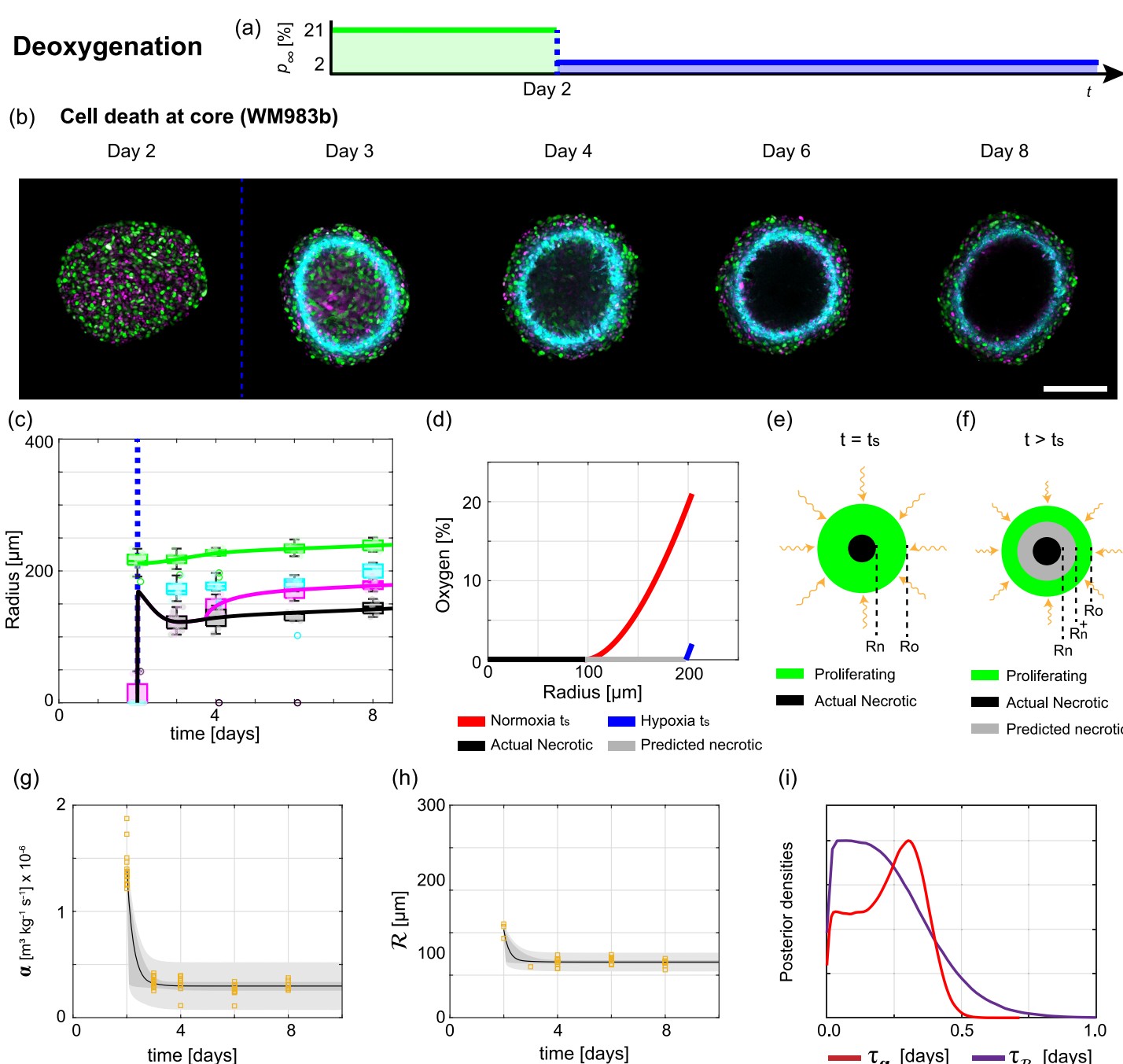

**Fig 4. Analysis of deoxygenation experiments reveals tumour spheroid adaptation mechanisms.** (a) Schematic for deoxygenation experiment where the external oxygen partial pressure switches from normoxia to hypoxia at $t_s = 2$ [days]. (b) Experimental images of the equatorial plane of WM983b spheroids on Days 2, 3, 4, 6, and 8 after seeding. Scale bar is 200 μm. Colours in (b) correspond to cell cycle schematic shown in Fig 1D and hypoxic regions are shown by pimonidazole staining (cyan). Necrotic region forms on Day 3 (blurry central region) and grows (dark central region on Days 4, 6, and 8). (c) Comparison of experimental data (box charts) with deoxygenation mathematical model (solid line) simulated with the maximum likelihood estimates of parameters. Time evolution of outer radius $R_o(t)$ (green), inhibited radius $R_i(t)$ (magenta), hypoxic radius $R_p(t)$ (cyan), and necrotic radius $R_n(t)$ (black). Blue vertical dashed lines in (b-c) indicate $t_s$. (d) Oxygen partial pressure within a spheroid estimated at $t_s$ under normoxia (red) and hypoxia (blue). (e-f) Schematics for spheroid structure in: (e) normoxia at $t = t_s$; and, (f) hypoxia immediately after $t_s$. In (f) the predicted necrotic radius, $R_n^+(t_s)$, is greater than the actual necrotic radius, $R_n(t_s)$, due to the reduced external oxygen partial pressure. Cells in region $R_n^+(t) < r < R_n(t)$ die due to lack of oxygen and increase the volume of the necrotic core. (g-i) Analysis of spheroid snapshots in deoxygenation experiment. (g) Estimates of oxygen consumption rate, $\alpha$ [m³ kg⁻¹ s⁻¹], from analysis of spheroid snapshots (orange squares) compared to prediction intervals generated from the deoxygenation mathematical model. (h) Estimates of the outer radius when the inhibited region forms, $\mathcal{R}(t)$ [μm], from analysis of spheroid snapshots (orange squares) compared to prediction intervals generated from the mathematical model. In (g-h) deoxygenation model simulated with the posterior means of the parameters (black), 50% posterior region of prediction interval (dark grey), and 97.5% posterior region of prediction interval (light grey). (i) Posterior density estimates for the timescales of adaptation of the oxygen consumption rate and outer radius when the inhibited region forms denoted by $\tau_\alpha$ (purple) and $\tau_\mathcal{R}$ (red).

$$s(t) = s_h + (s_n - s_h)\exp\left(-\frac{1}{\tau_s}(t - t_s)\right), \tag{4.3}$$

$$\mathcal{R}(t) = \mathcal{R}_h + (\mathcal{R}_n - \mathcal{R}_h)\exp\left(-\frac{1}{\tau_\mathcal{R}}(t - t_s)\right), \tag{4.4}$$

where $\tau_\alpha$ [days], $\tau_\lambda$ [days], $\tau_s$ [days], and $\tau_\mathcal{R}$ [days] denote timescales of adaptation for $\alpha$, $\lambda$, $s$, and $\mathcal{R}$, respectively. Further, the new constants in Eq (4) with subscripts $n$ and $h$, for example $\alpha_h$ and $\alpha_n$, represent parameter estimates from spheroids grown in normoxia and hypoxia, respectively. The other parameters ($k$, $\Omega$, $\kappa$), are assumed to be constants. Hence, $R_c^2(t) = 6kp_\infty/(\alpha(t)\Omega)$ [μm²], $Q^2(t) = \mathcal{R}^2(t)R_c^2(t)$ [-], and $\gamma(t) = \lambda(t)/s(t)$ [-] are functions of time.

In the region $R_n(t) < r < R_n^+(t)$ we assume cells die and increase the size of the necrotic core at a rate $\hat{\lambda}(t) = \hat{\lambda}\exp((t - t_s)/\tau_{\hat{\lambda}}) > 0$ [day$^{-1}$] per unit volume (Fig 4F). Conservation of volume for the necrotic core at time $t$, $V_n(t)$, gives (Text C.2.1 in S1 File)

$$\frac{dV_n(t)}{dt} = 3\hat{\lambda}(t)\left[\frac{4\pi}{3}R_n^+(t)^3 - V_n(t)\right] - 3\lambda(t)V_n(t). \tag{5}$$

Volume is converted to radius for comparison with experimental data using $R_n^3(t) = 3V_n(t)/4\pi$. At later times the term involving $\hat{\lambda}(t)$ dominates the right hand side of Eq (5) and $R_n(t)$ tends to $R_n^+(t)$. Eq (1), obtained by conservation of volume arguments, remains valid by including the time dependence in $s(t)$ and $\lambda(t)$. At $t_s$ there is no immediate change in the waste concentration within the spheroid and so no immediate change to $R_i(t)$. However, as $\mathcal{R}(t)$ changes with time (Eq (4.4)) the waste concentration within the spheroid and $R_i(t)$ evolve over time in part directly due to deoxygenation.

In this new mathematical model (Eqs (8.1)–(8.11)) there are fifteen parameters $\Theta_d = (R_o(0),$ $\alpha_n, \alpha_h, \tau_\alpha, \mathcal{R}_n, \mathcal{R}_h, \tau_\mathcal{R}, s_n, s_h, \tau_s, \lambda_n, \lambda_h, \tau_\lambda, \hat{\lambda}, \tau_{\hat{\lambda}})$. However, using Bayesian inference for parameter estimation the biological adaptation mechanisms become clearer (Methods: Parameter estimation and identifiability analysis). We identify fast adaptation to deoxygenation in $\alpha(t)$ with mean $\tau_\alpha = 0.23$ [days] and for $\mathcal{R}(t)$ with mean $\tau_\mathcal{R} = 0.23$ [days] (Fig 4G–4J). Furthermore, simulating the new deoxygenation mathematical model we find good agreement with the experimental measurements of $R_o(t)$, $R_n(t)$, and $R_i(t)$ (Fig 4C and Fig Z in S1 File). Therefore, our new mathematical model provides a mechanistic description to the observed growth dynamics in the variable external environment and appears to capture key adaptation mechanisms.

## Adaptation to re-oxygenation

We also perform re-oxygenation experiments, where spheroids grown in normoxia are transferred to hypoxic conditions at time $t_s$. These re-oxygenation experiments exhibit a range of unexpected biological adaptation mechanisms for each cell line that appear to depend on: $t_s$; spheroid size at re-oxygenation, $R_o(t_s)$; and necrotic core fraction at re-oxygenation, $\xi_n(t_s) = R_n(t_s)/R_o(t_s)$.

First we focus on slower growing WM793b spheroids and $t_s = 2$ [days] (Fig 5A and 5C). In hypoxic conditions prior to re-oxygenation, experimental images show a large hypoxic region (Day 2 in Fig 5A, 5C and 5G). However, after re-oxygenation at $t_s + 1$ [days] there is no hypoxic region (Day 3 in Fig 5C and 5G). Spheroid growth after deoxygenation appears to progress similar to spheroids that are grown in normoxia throughout (Days 3–8 in Fig 5C and 5G).

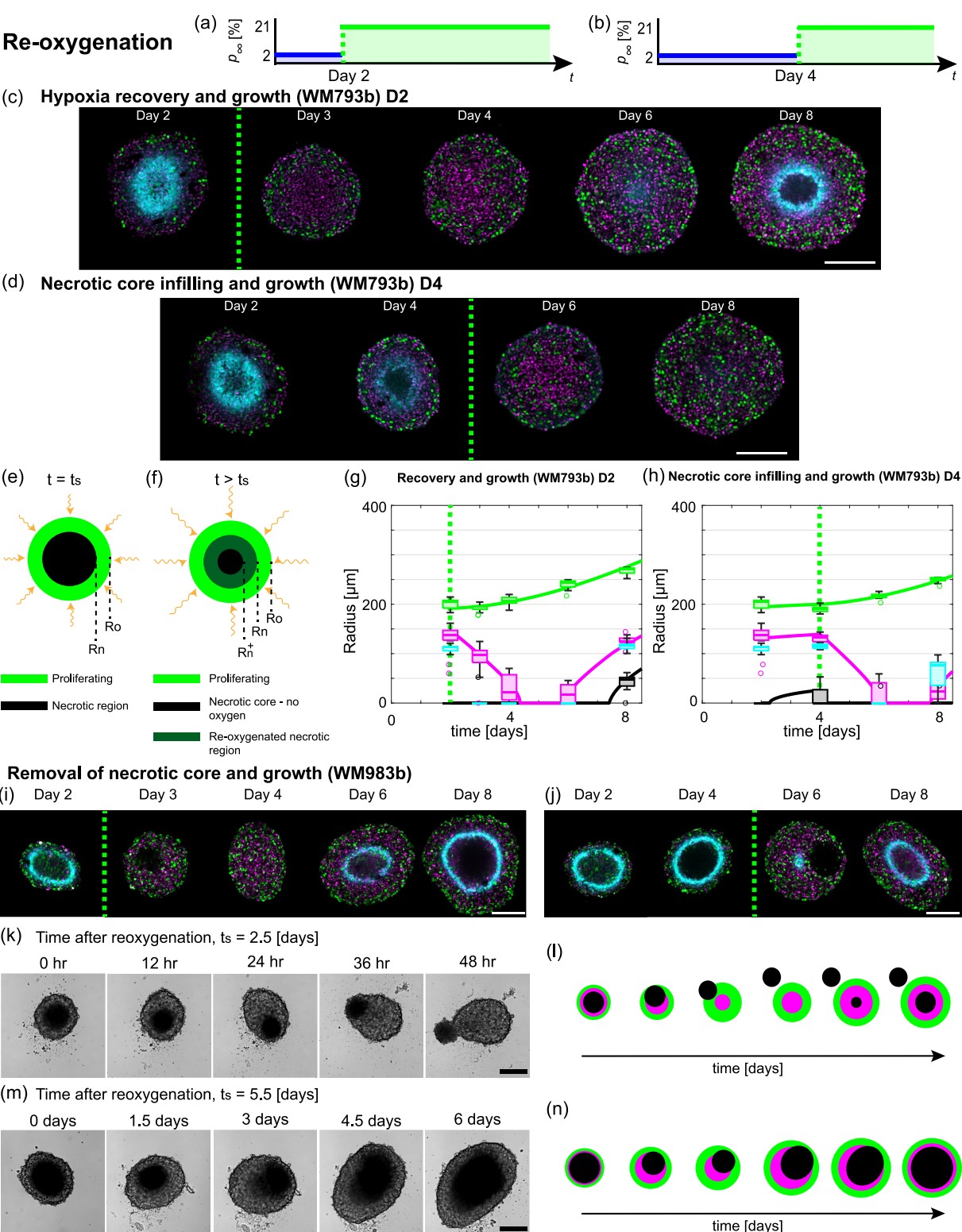

**Fig 5. Tumour spheroids exhibit a range of adaption mechanisms in response to re-oxygenation.** (a,b) Schematics for re-oxygenation experiments, where the external oxygen environment switches from hypoxia to normoxia at (a) $t_s = 2$ [days] and (b) $t_s = 4$ [days]. (c,g) Hypoxia recovery and growth for WM793b cell line with $t_s = 2$ [days] with (c) experimental images and (g) radial measurements. (d,h) Necrotic core infilling and growth for WM793b cell line with $t_s = 4$ [days] with (d) experimental images and (h) radial measurements. Green dashed line in (c, d,g,h,i,j) indicate $t_s$. Colours in (g,h) $R_o(t)$ (green), $R_i(t)$ (magenta), $R_n(t)$ (black), and $R_p(t)$ (cyan). (e,f) Schematics for tumour spheroid

structure due to re-oxygenation at $t_s$. (i-l) Removal of necrotic core and growth for WM983b cell line. (i) Confocal images for experiment with $t_s$ = 2 [days]. (j) Confocal images for experiment with $t_s$ = 4 [days]. (k) Bright-field images for experiment with $t_s$ = 2.5 [days]. (l) Schematic for removal of necrotic core and growth in WM983b cell line. (m) Bright-field images for experiment with $t_s$ = 5.5 [days]. (n) Schematic for movement of necrotic core and growth in WM983b cell line. Colours in (c,d,i,j) correspond to cell cycle schematic shown in Fig 1D and hypoxic regions are shown by pimonidazole staining (cyan). Scale bars in (c,d,i,j,k,m) are 200 μm.

For WM793b spheroids and $t_s$ = 4 [days], a necrotic core forms before $t_s$ (Day 4 in Fig 5D and 5H). However, after deoxygenation, at $t_s$ + 2 [days] there is no necrotic core (Day 6 in Fig 5D and 5H). At $t_s$ + 4 [days] spheroids are either in phase (i) or phase (ii) (Day 8 in Fig 5D and 5H). While traditional tumour spheroid experiments progress through phase (i), (ii), and (iii) sequentially, these experiments show that spheroids can transition transiently through the growth phases in reverse order before subsequently growing in the usual order.

To interpret the WM793b re-oxygenation experiments we proceed analogously to the deoxygenation experiments. We extend Greenspan's mathematical model to account for differences in parameter estimates between normoxia and hypoxia. Estimating the oxygen partial pressure within the spheroid at $t_s$, the predicted necrotic core, denoted $R_n^+(t_s)$, is smaller than the actual necrotic core, $R_n(t_s)$, provided $R_n(t_s) > 0$. In the region $R_n^+(t) < r < R_n(t)$ where the necrotic core is now supplied with oxygen, we assume that size of the necrotic core decreases at a rate $\tilde{\lambda}(t) = \tilde{\lambda}\exp((t - t_s)/\tau_{\tilde{\lambda}}) > 0$ [day$^{-1}$] per unit volume. We assume that a fraction, $0 \leq \nu \leq 1$, of the volume lost from the necrotic core recovers from the harsh oxygen conditions and increases the population of living cells, and the remaining volume lost from the necrotic core diffuses out of the spheroid and does not influence $R_i(t)$. Using Bayesian inference we estimate the seventeen model parameters, $\Theta_r = (R_o(0), R_n(0), \alpha_n, \alpha_h, \tau_\alpha, \mathcal{R}_n, \mathcal{R}_h, \tau_{\mathcal{R}}, s_n, s_h, \tau_s, \lambda_n, \lambda_h, \tau_\lambda, \tilde{\lambda}, \tau_{\tilde{\lambda}}, \nu)$ and simulate the mathematical model. We observe good agreement with the experimental data suggesting our new re-oxygenation mathematical model captures key mechanisms underlying adaptation and growth (Fig 5G and 5H, and Figs BB and CC in S1 File).

Results for WM983b spheroids are unexpected. We hypothesised, based on the exploration of the mathematical model, that spheroid growth dynamics may occur in reverse as observed for WM793b spheroids. However, we did not anticipate that WM983b spheroids would lose their symmetrical internal structure and necrotic core due to re-oxygenation (Fig 5I–5N, S1 and S2 Movies). The WM983b experiments are performed at four different re-oxygenation times $t_s$ = 2, 2.5, 4, and 5.5 [days] (Fig 5I, 5J, 5K and 5M). For all experiments, spheroids at $t_s$ prior to re-oxygenation are in phase (iii). For $t_s$ = 2, confocal microscopy reveals that at $t_s$ + 1 [days] there is a necrotic region but it is not at the centre of the spheroid (Day 3 of Fig 5I). At later times there is no necrotic region and growth proceeds analogous to spheroids in normoxia. Similarly, for $t_s$ = 4 (Fig 5J).

To explore this unusual behaviour for $t_s$ = 2.5 [days] and $t_s$ = 5.5 [days] we perform experiments using the IncuCyte S3 live cell imaging system (Sartorius, Goettingen, Germany) and obtain hourly bright-field images after re-oxygenation. In Fig 5K and S1 Movie with $t_s$ = 2.5 [days], initially the necrotic core of the spheroid is visible as a dark central region. At later times the necrotic core is located closer to the edge of the spheroid and the radially symmetric internal structure is lost (12, 24 and 36 hours after re-oxygenation in Fig 5K). At $t_s$ + 2 [days] the necrotic core appears to have exited the spheroid as a single object (48 hours after re-oxygenation in Fig 5K). Tracking the position of the necrotic core relative to the spheroid suggests the necrotic core moves randomly (Text D.3.1 in S1 File).

We observe similar behaviour for WM983b spheroids with $t_s$ = 5.5 [days] (Fig 5M and S2 Movie). However, likely due to the larger $R_o(t_s)$ and $\xi_n(t_s)$ here, the necrotic core is close to the edge of the spheroid but does not exit as a single object (1.5 and 3 days after $t_s$ in Fig 5M).

Instead as the spheroid grows necrotic matter forms at the centre of the spheroid and appears to merge with the necrotic matter located closer to the periphery (3, 4.5, and 6 days after $t_s$ in Fig 5M). As the WM983b spheroids do not maintain spherical symmetry we do not interpret these experimental data with the re-oxygenation mathematical model. Schematics describing the behaviours are presented in Fig 5L and 5N.

Confocal images for WM164 spheroids suggest that symmetry is maintained (Figs P and Q in S1 File), and that WM164 spheroids transition transiently through the growth phases in reverse order before subsequently growing in the usual order. In contrast to WM793b spheroids, WM164 spheroids can resume growth while maintaining a necrotic core (Fig CC in S1 File). These different behaviours appear to be dependent on $t_s$, $R_o(t_s)$ and $\xi_n(t_s)$. To interpret WM164 re-oxygenation experiments we use the re-oxygenation model (Text F in S1 File). However, additional care should be exercised interpreting WM164 re-oxygenation results. Brightfield time-lapse images show that mass from the necrotic core can move to the periphery and exit the spheroid (Fig R in S1 File and S3 Movie).

## Discussion

Tumours grow in complicated fluctuating external environments. However, spheroid experiments used to study tumours are typically performed in constant external environments and with oxygen partial pressures that are much greater than *in vivo*. To explore this gap between *in vitro* and *in vivo* conditions, we analyse a series of tumour spheroid experiments using mathematical modelling and statistical uncertainty quantification. Growing spheroids in time-dependent external oxygen conditions reveals a range of behaviours not observed in standard experimental protocols. For fifty years, tumour spheroid growth has been characterised by three sequential growth phases: phase (i) exponential growth; phase (ii) reduced exponential growth; and phase (iii) saturation. However, here in re-oxygenation experiments, spheroids can transiently undergo these growth phases in reverse. Furthermore, spheroids can lose their spherically symmetric structure and necrotic core. Deoxygenation experiments also show that large changes to the internal structure of spheroids can occur while the overall size remains constant. Overall, our results suggest that oxygen and internal structure play pivotal roles in spheroid growth and should be taken into account when interpreting spheroid experiments. This is important as many studies do not provide sufficient information to replicate oxygen conditions and do not measure spheroid internal structures.

Tumour spheroid growth is a complex process involving multiple mechanisms. However, the contribution of each mechanism to growth is unclear using experimentation alone. To quantitatively explore which mechanisms contribute to spheroid growth we use the seminal Greenspan mathematical model [10]. The model describes the growth of spheroids in normoxia and hypoxia remarkably well. Moreover, our analysis suggests that growth and formation of the necrotic core is reasonably described by oxygen diffusion and consumption, whereas the growth and formation of the inhibited region is more accurately described by waste production and diffusion. Using statistical uncertainty quantification we show that the rates at which different biological processes occur differ between normoxia and hypoxia. Therefore, external environmental conditions should be taken into account when interpreting tumour spheroid experiments. Previous studies analysing previously available experimental data with Greenspan's model [15, 16] have not been able to distinguish between the two mechanisms. Further, by considering the temporal evolution of spheroid structure, our results build on studies analysing spheroid snapshots only [17, 37]. As the model captures the key dynamics, we do not include other biological mechanisms that may be relevant in future studies, for example glucose [4]. Introducing additional mechanisms prior to developing the

deoxygenation and re-oxygenation models would complicate the analysis, and likely result in parameters being non-identifiable and not physically interpretable. Both of which we aim to avoid. Many mathematical models have been developed with additional mechanisms, but they have not been quantitatively tested with experimental data [25, 31–34]. The experimental data and framework that we provide here are suitable to test such models.

Deoxygenation and re-oxygenation experiments reveal how spheroid overall size and internal structure adapt to changes in the external environment. However, without a mathematical modelling and statistical uncertainty quantification framework like we use here, the mechanisms underlying adaptation are challenging to identify and interpret. Extending Greenspan's mathematical model allows us to interpret, analyse, and describe these deoxygenation and re-oxygenation experiments remarkably well. Parameter estimation and identifiability analysis, using profile likelihood analysis and Bayesian inference, allows us to identify and quantify the contributions of key biological mechanisms to adaptation and growth. For both models, the analysis identifies a narrow range of behaviours that differ in terms of the rates of adaptation in $t_s < t < t_s + 1$ [days] and long term dynamics (Text D.2.1 in S1 File). These modelling predictions raise interesting questions (Text D.2.1 in S1 File). In comparison to standard experimental protocols, we collect more data per time point and over a longer duration. Further, we build on previous studies [13, 15, 16] to improve on standard experimental designs by measuring the internal structure of spheroids and hypoxic regions in addition to the overall size. Even these improvements to standard protocols are insufficient to identify all adaptation processes. As with all studies, additional data would be beneficial. In particular, frequent measurements of oxygen and internal structure at early times would be useful but are challenging to obtain experimentally (Text D.2 in S1 File).

Re-oxygenation experiments reveal unexpected necrotic core removal in WM983b spheroids. To the best of our knowledge this behaviour has not been previously described. The exact mechanisms underlying this behaviour are unclear. We hypothesise that small asymmetry at re-oxygenation, possibly in the distribution of proliferating cells, in combination with changes to cell-cell adhesion and physical interactions contribute. We also observe that mass from the necrotic core of WM164 spheroids can move to the periphery and exit the spheroid. It is unclear whether these observations are relevant to other cell lines and to *in situ* tumours. Interesting future work is to explore these unusual behaviours in greater detail experimentally. For example, can this phenomenon be induced by other external environmental changes, drug treatments, and *in vivo*. As the necrotic core removal destroys the spherical symmetry of the spheroid we do not use Greenspan's mathematical model to analyse these observations. It would be interesting future work to ascertain the physical and biological mechanisms that could explain these observations, possibly by extending a multiphase model [45, 46], studies on cellular blebbing [47], compound droplet models [48], or models with different treatments of necrotic material [46].

This work lays the foundation for further studies bridging the gap between clinical conditions and standard experimental protocols. Here, we consider normoxia and hypoxia and switches between normoxia and hypoxia. The impact of other oxygen conditions on spheroid growth and cell cycle progression within growing spheroids are also worth consideration, for example to mimic *in vivo* oxygen gradients [49], *in vivo* vascularisation, disrupted oxygen supplies, or cyclic hypoxia [1, 22]. Microfluidic devices may be one useful approach [25], but challenges visualising the internal structure of spheroids throughout such experiments must be overcome. Intracellular responses to oxygen changes are also of interest [50, 51]. Results in this study also depend upon Greenspan's mathematical model, and therefore assumptions of spherical symmetry. Interesting future work would be to explore relaxing spherical symmetry assumptions. Further interesting work would be to repeat the experiments using the EF5

hypoxia marker and compare results, repeat the experiments with different cell lines, explore similarities and differences with *in situ* tumours, and explore alternative mechanisms that might explain why estimates of $p_i$ differ across oxygen conditions. Alternative mathematical models to explain these experiments and provide further useful insights and predictions may also be of interest. For example, one could explore replacing the time-dependent adaptation mechanisms with functional forms incorporating additional biological mechanisms. When developing such alternative models one should take into account challenges with parameter identifiability that can arise as model complexity increases [38]. We also note that our framework is well suited to explore the role of other changing external conditions on spheroid growth, for example nutrient availability and mechanical confinement [4, 11], and can be extended to explore different treatment strategies, for example radiotherapy and chemotherapy [34, 41].

## Materials and methods

### Mathematical modelling

**Greenspan's mathematical model.** Key elements of Greenspan's mathematical model for fixed $p_\infty$ are included in the main text. Further details of the model derivation are included in Text C.1 in S1 File. Recall that this model has two interpretations, that we refer to as hypotheses 1 and 2.

- Hypothesis 1 assumes that the necrotic and inhibited regions are both driven by oxygen diffusion and consumption.

- Hypothesis 2 assumes that the necrotic region is driven by oxygen diffusion and consumption whereas the inhibited region is driven by waste production and diffusion.

Here, we present the governing differential-algebraic system of equations for the outer radius, $R_o(t)$, necrotic radius, $R_n(t)$, and inhibited radius, $R_i(t)$, for both hypotheses 1 and 2,

$$R_o^2(t) \frac{dR_o(t)}{dt} = \frac{s}{3} \left[ R_o^3(t) - \max\left( R_i(t)^3, R_n^3(t) \right) \right] - \lambda R_n(t)^3, \tag{6.1}$$

$$R_c^2 = R_o^2(t) - R_n^2(t) - \frac{2R_n^2(t)}{R_o(t)} (R_o(t) - R_n(t)), \tag{6.2}$$

$$\mathcal{R}^2 = R_o^2(t) - R_i^2(t) - 2R_n^3(t) \left( \frac{1}{R_i(t)} - \frac{1}{R_o(t)} \right). \tag{6.3}$$

In these experiments, Eq (6.1) simplifies as $R_i(t) \geq R_n(t)$, consistent with our parameter choices (Methods: Parameter estimation and identifiability analysis) [16]. For both hypothesis 1 and 2: the outer radius when the necrotic region forms is $R_c = [6kp_\infty/(\alpha\Omega)]^{1/2}$; Eq (6.1) arises from conservation of volume; and, Eq (6.2) is obtained by evaluating the oxygen partial pressure within the spheroid at the necrotic threshold. For hypothesis 1, Eq (6.3) is obtained by evaluating the oxygen partial pressure within the spheroid at the oxygen inhibited threshold, $p_i$, and $Q^2 = (p_\infty - p_i)/p_\infty$ so the left hand side of Eq (6.3) is $\mathcal{R}^2 = R_c^2 Q^2 = 6k(p_\infty - p_i)/(\alpha\Omega)$. In contrast, for hypothesis 2, Eq (6.3) arises by evaluating the waste concentration within the spheroid at the waste inhibited threshold, $\beta_i$, and $Q^2 = \beta_i\kappa\alpha\Omega/(Pkp_\infty)$ so the left hand side of Eq (6.3) is $\mathcal{R}^2 = R_c^2 Q^2 = 6\beta_i/P$. We solve the system of Eqs (6.1)–(6.3) numerically (Text C.1.2 in S1 File).

Analysing the model, the inhibited region forms at [10]

$$t = \frac{3}{s} \log\left(\frac{\mathcal{R}}{R_o(0)}\right).$$ (7.1)

For hypothesis 1, Eq (7.1) is $t = (3/s) \log([6k(p_\infty - p_i)/\alpha]^{1/2}/R_o(0))$. For hypothesis, 2 Eq (7.1) is $t = (3/s) \log([6\beta_i\kappa/P]^{1/2}/R_o(0))$.

**Mathematical model to interpret deoxygenation experiments.** Key elements of the mathematical model derived to interpret deoxygenation experiments are included in the main text. Further details of the model derivation are included in Text C.2.2 in S1 File. Here, we present the governing equations for $0 < t < t_s$ and $t > t_s$.

For $0 < t < t_s$ we solve Greenspan's mathematical model [10] in normoxia interpreting the governing mechanisms with hypothesis 2. The differential-algebraic system of Eqs (6.1)–(6.3) are solved to determine $R_o(t)$, $R_n(t)$, and $R_i(t)$.

After deoxygenation, $t > t_s$, we extend Greenspan's mathematical model to account for adaptation to hypoxia. Rewriting Eqs (4), (5) and (6.1), and solving Eq (6.2) for $R_n^+(t)$ instead of $R_n(t)$, gives the governing differential-algebraic system of equations

$$R_o^2(t)\frac{dR_o(t)}{dt} = \frac{s(t)}{3}\left[R_o^3(t) - \max\left(R_i^3(t), R_n^3(t)\right)\right] - \lambda(t)R_n^3(t),$$ (8.1)

$$\frac{dV_n(t)}{dt} = 3\hat{\lambda}(t)\left[\frac{4\pi}{3}R_n^+(t)^3 - V_n(t)\right] - 3\lambda(t)V_n(t),$$ (8.2)

$$R_c^2(t) = R_o^2(t) - R_n^+(t)^2 - \frac{2R_n^+(t)^2}{R_o(t)}\left(R_o(t) - R_n^+(t)\right),$$ (8.3)

$$\mathcal{R}^2(t) = R_o^2(t) - R_i^2(t) - 2R_n^3(t)\left(\frac{1}{R_i(t)} - \frac{1}{R_o(t)}\right),$$ (8.4)

$$\alpha(t) = \alpha_h + (\alpha_n - \alpha_h)\exp\left(-\frac{1}{\tau_\alpha}(t - t_s)\right),$$ (8.5)

$$\lambda(t) = \lambda_h + (\lambda_n - \lambda_h)\exp\left(-\frac{1}{\tau_\lambda}(t - t_s)\right),$$ (8.6)

$$s(t) = s_h + (s_n - s_h)\exp\left(-\frac{1}{\tau_s}(t - t_s)\right),$$ (8.7)

$$\mathcal{R}(t) = \mathcal{R}_h + (\mathcal{R}_n - \mathcal{R}_h)\exp\left(-\frac{1}{\tau_\mathcal{R}}(t - t_s)\right),$$ (8.8)

$$\hat{\lambda}(t) = \hat{\lambda}\exp\left(\frac{1}{\tau_{\hat{\lambda}}}(t - t_s)\right),$$ (8.9)

$$R_n(t) = \left[\frac{3}{4\pi}V_n(t)\right]^{\frac{1}{3}},$$ (8.10)

$$R_c^2(t) = \frac{6kp_\infty}{\alpha(t)\Omega}. \tag{8.11}$$

Note that in the long time limit $t \to \infty$, we recover Greenspan's mathematical model for normoxia (Eqs (6.1)–(6.3)). Specifically, $\alpha(t) \to \alpha_h$, $\lambda(t) \to \lambda_h$, $s(t) \to s_h$ and $\mathcal{R}(t) \to \mathcal{R}_h$ as $t \to \infty$. Further, the term involving $\hat{\lambda}(t)$ dominates the right hand side of Eq (8.2) as $t \to \infty$, so $R_n(t) \to R_n^+(t)$ as $t \to \infty$. We solve the system of Eqs (8.1)–(8.11) numerically (Text C.2.2 in S1 File).

**Mathematical model to interpret re-oxygenation experiments.** Key details of the mathematical model to interpret re-oxygenation experiments are included in Text C.3 in S1 File.

## Parameter estimation and identifiability analysis

Parameter estimation and identifiability analysis are performed using profile likelihood analysis [15, 16, 52, 53], Bayesian inference [54–56], and global optimisation techniques, as now detailed. Throughout we exclude outliers in the experimental data. To detect outliers for each experiment we analyse each measurement type, $R_o(t)$, $R_n(t)$, and $R_i(t)$ at each time point independently using MATLABs isoutlier function with method *quartiles*.

**Greenspan's model.** Parameter estimation and identifiability analysis for Greenspan's model are first performed using profile likelihood analysis (Fig 3), see [16]. The Bayesian inference approach to estimate parameters of Greenspan's model is discussed in the following.

**Deoxygenation experiments.** Parameter estimation for the deoxygenation model (Eqs (8.1)–(8.11)) is performed using global optimisation and Bayesian inference techniques. We now explain how we estimate the fifteen model parameters, $\Theta_d = (R_o(0), \alpha_n, \alpha_h, \tau_\alpha, \mathcal{R}_n, \mathcal{R}_h, \tau_\mathcal{R}, s_n, s_h, \tau_s, \lambda_n, \lambda_h, \tau_\lambda, \hat{\lambda}, \tau_{\hat{\lambda}})$. Informed by experimental measurements we set $R_n(t_s) = 0$ and $R_i(t_s) = 0$, but they could be included as additional parameters in future work.

We revisit Greenspan's mathematical model for spheroids grown in normoxia and hypoxia for first estimates of $s_n$, $s_h$, $\lambda_n$ and $\lambda_h$. Starting with experimental data from spheroids grown in normoxia, we fit a normal distribution to the initial outer radius measurements using the MATLAB fitdist function. Then for each spheroid we estimate $R_c$ using Eq (6.2) given measurements of $R_o(t)$ and $R_n(t)$ and fit a normal distribution using the MATLAB fitdist function. Similarly, we estimate $\mathcal{R}$ using Eq (6.3) and measurements of $R_o(t)$, $R_n(t)$ and $R_i(t)$ and fit a normal distribution using the MATLAB fitdist function. Next, we seek to estimate the five parameters of Greenspan's model, $\Theta_n = (R_o(0), R_c, s, \lambda, \mathcal{R})$, using the MATLAB package *MCMCstat* developed by Marko Laine [57, 58]. Detailed information on the *MCMCstat* package is available on the GitHub repository (https://mjlaine.github.io/mcmcstat/).

Before we can use the *MCMCstat* package we require good first estimates of $\Theta_n$ and the mean squared error, *mse*. To provide a good first estimate for $\Theta_n$, we perform global optimisation using the MATLAB GlobalSearch function with settings: fmincon *sqp* algorithm; *MaxTime* = 60 [minutes]; *NumTrialPoints* = 5000; and *lowerbounds* and *upperbounds* informed by fitted normal distributions for $R_o(0)$, $R_c$, and $\mathcal{R}$ and previous results [16]. To estimate the *mse* we use experimental measurements as observations and simulate the deoxygenation model with the estimate of $\Theta_n$ from global optimisation to obtain predicted values. Next we use the *MCMCstat* package to generate four MCMC chains with 250,000 samples and enable automatic sampling and estimation of the error standard deviation. Next we discard the first 50,000 samples of each of the four chains as burn-in, resulting in each chain containing 200,000 samples. For other *MCMCstat* package options we use the default settings. To assess

convergence of the MCMC chains we compute the potential scale reduction factor, $\hat{R}$, [56] for each parameter and observe convergence, where convergence corresponds to $\hat{R} < 1.1$ (Table B in S1 File). Performing posterior checks, using 50,000 samples from the chains to generate prediction intervals and comparing with the experimental data, suggests the parameter estimates are reasonable. This process is repeated for the hypoxia data to estimate $\Theta_h$ (Table C in S1 File). The posteriors generated here are for $s_n$, $s_h$, $\lambda_n$ and $\lambda_h$.

Next, we analyse the deoxygenation experimental data at each spheroid and time point independently. To estimate $\mathcal{R}_n$, $\mathcal{R}_h$, $\tau_{\mathcal{R}}$ from Eq (8.8) we use global minimisation and the *MCMCstat* package. For estimates of $\mathcal{R}$ for each spheroid and at each time point independently we use Eq (6.2) and measurements of $R_o(t)$ and $R_n(t)$. Similarly, to estimate $\alpha_n$, $\alpha_h$, $\tau_{\alpha}$ from Eq (8.5) we use global minimisation and the *MCMCstat* package. For estimates of $\alpha$ we use Eqs (8.11) and (6.3) and measurements of $R_o(t)$, $R_n(t)$ and $R_i(t)$. Note that to estimate $R_c$ we assume that $R_n(t) = R_n^+(t)$. To estimate $R_o(0)$ we fit a normal distribution using the MATLAB `fitdist` function to measurements of $R_o(t)$ at the first time point.

Next we perform a global minimisation to estimate $\Theta_d$, using the estimates of $R_o(0)$, $\alpha_n$, $\alpha_h$, $\tau_{\alpha}$, $\mathcal{R}_n$, $\mathcal{R}_h$, $\tau_{\mathcal{R}}$, $s_n$, $s_h$, $\lambda_n$, $\lambda_h$ to inform *lowerbounds* and *upperbounds*. To estimate $\Theta_d$ we use global minimisation and the *MCMCstat* package (Table D In S1 File). Performing posterior checks, using 50,000 samples from the chain to generate prediction intervals and comparing with the experimental data, suggests the parameter estimates are reasonable (Figs Z, BB, and CC in S1 File).

**Re-oxygenation experiments.** Parameter estimation for the re-oxygenation model formed by Eqs (S.36.1)-(S.36.11) in Text C.3 of S1 File, with parameters $\Theta_r = (R_o(0), R_n(0), \alpha_n, \alpha_h, \tau_{\alpha}, \mathcal{R}_n, \mathcal{R}_h, \tau_{\mathcal{R}}, s_n, s_h, \tau_s, \lambda_n, \lambda_h, \tau_{\lambda}, \tilde{\lambda}, \tau_{\tilde{\lambda}}, \nu)$, is analogous to the approach used for the deoxygenation model. Summary statistics for the MCMC chains and MCMC diagnostics are shown in Tables E and F in S1 File. Comparing prediction intervals and with the experimental data suggests the parameter estimates are reasonable (Figs BB and CC in S1 File).

## Experimental methods

*Cell culture.* The human melanoma cell lines established from primary (WM793b) and metastatic cancer sites (WM983b, WM164) were provided by Prof. Meenhard Herlyn, The Wistar Institute, Philadelphia, PA, [59]. All cell lines were previously transduced with fluorescent ubiquitination-based cell cycle indicator (FUCCI) constructs [13, 14]. Cell lines were previously genotypically characterised [13, 60–62], and authenticated by short tandem repeat fingerprinting (QIMR Berghofer Medical Research Institute, Herston, Australia). The cells were cultured in melanoma cell medium ("Tu4% medium"): 80% MCDB-153 medium (Sigma-Aldrich, M7403), 20% L-15 medium (Sigma-Aldrich, L1518), 4% fetal bovine serum (ThermoFisher Scientific, 25080–094), 5 mg ml$^{-1}$ insulin (Sigma-Aldrich, I0516), 1.68 mM CaCl$_2$ (Sigma-Aldrich, 5670) [14]. Cell lines were checked routinely for mycoplasma and tested negative using the MycoAlert MycoPlasma Detection Kit (Lonza) and polymerase chain reaction [63].

*Spheroid generation, culture, and experiments.* Spheroids were generated in 96-well cell culture flat-bottomed plates (3599, Corning), with 5000 total cells/well, using 50 µl total/well non-adherent 1.5% agarose to promote formation of a single spheroid per well [30]. From previous work we expect that different seeding densities, in the range 1250–10000 total cells/well, will provide similar results [15, 16]. For all experiments spheroids formed after 2 days for WM793b, WM164 and WM983b. On day 3 and 7 of each experiment 50% of the medium in each well was replaced with fresh medium (200 µl total/well). Each experiment was performed for 8 days. This choice of experimental duration is informed by previous experiments with

these cell lines so that necrotic and inhibited region form prior to the end of the experiments and so that parameters of Greenspan's model are practically identifiable and can be estimated [16].

For normoxia experiments, cells and spheroids were grown and formed in an incubator with standard settings: 37˚C, 5% $CO_2$ [13, 14]; referred to as the normoxia incubator. For hypoxia experiments, cells were cultured in the normoxia incubator and then spheroids were grown in a hypoxia incubator with settings: 37˚C, 5% $CO_2$, 2% $O_2$. For deoxygenation experiments, cells and spheroids were formed and grown in the normoxia incubator. At the time of deoxygenation, the relevant plate(s) of spheroids were manually transferred from the normoxia incubator to the hypoxia incubator. Similarly, for the re-oxygenation experiments spheroids were grown in the hypoxia incubator then at the time of re-oxygenation the relevant plate(s) of spheroids were manually transferred to the normoxia incubator. The time to move plates was 1–2 minutes.

To estimate the outer, necrotic, inhibited, and hypoxic radii, we use a high-throughput method of mounting, clearing and imaging [64]. Results using this optical sectioning method are consistent with different methods such as cryosectioning [64]. Spheroids maintained in the relevant incubator were harvested, fixed with 4% paraformaldehyde (PFA), and stored in phosphate buffered saline solution (PBS), sodium azide (0.02%), Tween-20 (0.1%), and DAPI (1:2500) at 4˚C, on days 2, 3, 4, 6 and 8 after seeding. For all hypoxia measurements, spheroids were stained with 100 mM pimonidazole for three hours, prior to fixation. Spheroids were then permeabilized with 0.5% Triton X-100 in PBS for one hour; blocked in antibody dilution buffer (Abdil) [65] for 24 hours; stained with a 1:50 anti-pimonidazole mouse IgG1 monoclonal antibody (Hypoxyprobe-1 MAb1) in Abdil for 48 hours; washed in PBS with 0.1% Tween-20 for 6; placed in a 1:100 solution of goat anti-mouse Alexa Flour 647 in Abdil for 48 hours; and, finally washed for 6 hours in PBS. Then for imaging, fixed spheroids were set in place using low melting 2% agarose and optically cleared in 500 μl total/well high refractive index mounting solution (Quadrol 9% wt/wt, Urea 22% wt/wt, Sucrose 44% wt/wt, Triton X-100 0.1% wt/wt, water) for 2 days in a 24-well glass bottom plate (Cellvis, P24–1.5H-N) before imaging to ensure accurate measurements [64, 66, 67]. Images were then captured using an Olympus FV3000 confocal microscope with the 10 × objective focused on the equatorial plane of each spheroid.

As the unexpected necrotic core movement for WM983b cell line was observed in the re-oxygenation experiments, the re-oxygenation experiments were repeated for all cell lines alongside a control normoxia condition. Spheroids were cultured into three 96-well plates (3599, Corning): plate (i) control for normoxia; plate (ii) re-oxygenation 2.5 days after seeding; and, plate (iii) re-oxygenation 5.5 days after seeding. Each plate consisted of 32 spheroids of each cell line. The plates were placed inside the IncuCyte S3 live cell imaging system (Sartorius, Goettingen, Germany) incubator (37˚C, 5% $CO_2$). IncuCyte S3 settings were chosen to image with the 10 × objective. For plate (i) images were captured every 2 hours for the first three days and then every 4 hours for the remainder of the experiment. For plate (ii) and (iii) images were captured every hour for three and seven days, respectively.

*Image processing.*Confocal microscopy images were converted to TIFF files in ImageJ and then processed with custom MATLAB scripts that use standard MATLAB image processing toolbox functions. Area was converted to an equivalent radius ($r^2 = A/\pi$). These scripts are freely available on Zenodo with DOI:10.5281/zenodo.5121093 [68], with modifications to account for pimonidazole staining and blurred central regions due to hypoxia and deoxygenation discussed in Text B in S1 File. Images captured with the IncuCyte S3 were processed with custom MATLAB scripts that use standard MATLAB image processing toolbox functions

(Text D.3.1 in S1 File). Note that we identify the boundary between the inhibited and proliferating region to be the radial position at which 20% of cells are cycling.

*Statistics and Reproducibility.* Details of practical parameter identifiability analysis and the Bayesian inference are presented in Methods: Parameter estimation and identifiability analysis. Each radial measurements is represented as an individual data point in relevant figures, with non-filled circles representing outliers (Methods: Parameter estimation and identifiability analysis), and are summarised using box charts. Table A in S1 File details the number of measurements at each time point for each cell line and experimental data analysed during the study are available on a GitHub repository (https://github.com/ryanmurphy42/Murphy2022SpheroidOxygenAdaptation). We note that some measurements could not be obtained primarily due to blurring of the automated imaging, spheroids not forming properly, or spheroids losing their structural integrity at late times. Data for these spheroids was excluded. In a previous study we assess experimental designs [16] and use this to inform that our sample size is sufficient in this study. Randomisation and blinding was not possible.

## Supporting information

**S1 File. Additional data and results.** This file includes: a summary of experimental data; experimental images; additional details of image processing; additional details of mathematical modelling; additional results for WM983b spheroids; additional results for WM793b cell line; additional results for WM164 cell line; and summary statistics of MCMC chains and MCMC diagnostics.

**Fig A: Experimental images of WM983b tumour spheroids in Experiment 1 - Normoxia.** Top set of images shows spheroids with FUCCI signal only. Bottom set of images show spheroids with FUCCI signal and pimonidazole staining. Scale bars are 400μm.

**Fig B: Experimental images of WM983b tumour spheroids in Experiment 2 - hypoxia.** Top set of images shows spheroids with FUCCI signal only. Bottom set of images show spheroids with FUCCI signal and pimonidazole staining. Scale bars are 400μm.

**Fig C: Experimental images of WM983b tumour spheroids in Experiment 3 - deoxygenation at $t_s$ = 2 [days] (blue dashed line).** Top set of images shows spheroids with FUCCI signal only. Bottom set of images show spheroids with FUCCI signal and pimonidazole staining. Scale bars are 400μm.

**Fig D: Experimental images of WM983b tumour spheroids in Experiment 6 - Re-oxygenation at $t_s$ = 2.5 [days].** Scale bars are 400μm.

**Fig E: Experimental images of WM983b tumour spheroids in Experiment 7 - Re-oxygenation at $t_s$ = 5.5 [days].** Scale bars are 400μm.

**Fig F: Experimental images of WM793b tumour spheroids in Experiment 1 - Normoxia.** Top set of images shows spheroids with FUCCI signal only. Bottom set of images show spheroids with FUCCI signal and pimonidazole staining. Scale bars are 400μm.

**Fig G: Experimental images of WM793b tumour spheroids in Experiment 2 - hypoxia.** Top set of images shows spheroids with FUCCI signal only. Bottom set of images show spheroids with FUCCI signal and pimonidazole staining. Scale bars are 400μm.

**Fig H: Experimental images of WM793b tumour spheroids in Experiment 3 - deoxygenation at $t_s$ = 2 [days] (blue dashed line).** Top set of images shows spheroids with FUCCI signal only. Bottom set of images show spheroids with FUCCI signal and pimonidazole staining. Scale bars are 400μm.

**Fig I: Experimental images of WM793b tumour spheroids in Experiment 4 - Re-oxygenation at $t_s$ = 2 [days] (green dashed line).** Top set of images shows spheroids with FUCCI signal only. Bottom set of images show spheroids with FUCCI signal and pimonidazole staining. Scale bars are 400μm.

**Fig J: Experimental images of WM793b tumour spheroids in Experiment 5 - Re-oxygenation at $t_s$ = 4 [days] (green dashed line).** Top set of images shows spheroids with FUCCI signal only. Bottom set of images show spheroids with FUCCI signal and pimonidazole staining. Scale bars are 400μm.

**Fig K: Experimental images of WM793b tumour spheroids in Experiment 6 - Re-oxygenation at $t_s$ = 2.5 [days].** Scale bars are 400μm.

**Fig L: Experimental images of WM793b tumour spheroids in Experiment 7 - Re-oxygenation at $t_s$ = 5.5 [days].** Scale bars are 400μm.

**Fig M: Experimental images of WM164 tumour spheroids in Experiment 1 - Normoxia.** Top set of images shows spheroids with FUCCI signal only. Bottom set of images show spheroids with FUCCI signal and pimonidazole staining. Scale bars are 400μm.

**Fig N: Experimental images of WM164 tumour spheroids in Experiment 2 - hypoxia.** Top set of images shows spheroids with FUCCI signal only. Bottom set of images show spheroids with FUCCI signal and pimonidazole staining. Scale bars are 400μm.

**Fig O: Experimental images of WM164 tumour spheroids in Experiment 3 - deoxygenation at $t_s$ = 2 [days] (blue dashed line).** Top set of images shows spheroids with FUCCI signal only. Bottom set of images show spheroids with FUCCI signal and pimonidazole staining. Scale bars are 400μm.

**Fig P: Experimental images of WM164 tumour spheroids in Experiment 4 - Re-oxygenation at $t_s$ = 2 [days] (green dashed line).** Top set of images shows spheroids with FUCCI signal only. Bottom set of images show spheroids with FUCCI signal and pimonidazole staining. Scale bars are 400μm.

**Fig Q: Experimental images of WM164 tumour spheroids in Experiment 5 - Re-oxygenation at $t_s$ = 4 [days] (green dashed line).** Top set of images shows spheroids with FUCCI signal only. Bottom set of images show spheroids with FUCCI signal and pimonidazole staining. Scale bars are 400μm.

**Fig R: Experimental images of WM164 tumour spheroids in Experiment 6 - Re-oxygenation at $t_s$ = 2.5 [days].** Scale bars are 400μm.

**Fig S: Experimental images of WM164 tumour spheroids in Experiment 7 - Re-oxygenation at $t_s$ = 5.5 [days].** Scale bars are 400μm.

**Fig T: Image processing to estimate $R_n(t)$ when the FUCCI signal in the central region of the spheroid is blurred.** Images show one WM983b spheroid one day after deoxygenation. Scale bars are 200 μm. (a) Spheroid with FUCCI signals (green and magenta). (b) Spheroid with FUCCI signals and pimonidazole staining (cyan). (c-e) Blurred region identified using the *polygon section* tool in ImageJ. Spheroid with: (c) FUCCI-green signal only (green), (d) FUCCI-red signal only (magenta), and (e) pimonidazole staining only (cyan). (f-g) Images in (a) and (b) with blurred region removed.

**Fig U: Additional experimental measurements comparing spheroids grown in normoxia and hypoxia.** (a-b) Estimates of $\xi_n(t) = R_n(t)/R_o(t)$. (a-b) Estimates of $\xi_n(t) = R_n(t)/R_o(t)$ with time for (a) normoxia and (b) hypoxia. (c-d) Estimates of $\xi_i(t) = R_i(t)/R_o(t)$ with time for (c) normoxia and (d) hypoxia. (e-f) Estimates of $\xi_p(t) = R_p(t)/R_o(t)$ with time for (e) hypoxia and (f) hypoxia. (g-h) Estimates of (g) $\xi_n(t)$, (h) $\xi_i(t)$, and (i) $\xi_p(t)$ with against outer radius, $R_o(t)$.

**Fig V: Oxygen diffusion and consumption is sufficient to describe formation and growth of necrotic core but not sufficient to describe formation and growth of inhibited region.** Estimates of $R_c$ (a) per spheroid; (b) with time; (c) against $R_o(t)$. Estimates of $\alpha$ (d) per spheroid; (e) with time; (f) against $R_o(t)$. Estimates of $p_i$ (g) per spheroid; (h) with time; (i) against $R_o(t)$. In (a-i) each data point represents a single spheroid. Data points are only included for $p_i$ if the spheroid is in phase (ii) or phase (iii), consistent with when equations used to estimate $p_i$ are valid.

**Fig W: Analysing whether production and diffusion of waste describes formation and growth of inhibited region.** Estimates of the outer radius when the inhibited region first forms, $\mathcal{R} = (\beta_i \kappa / P)^{1/2}$ (a) box chart; (b) per spheroid; (c) with time; (d) against $R_o(t)$. In (b-d) each data point represents a single spheroid. Data points are only included only if the spheroid is in phase (ii) or phase (iii), consistent with when the equations used to estimate $\mathcal{R}$ are valid.

**Fig X: Bayesian inference to estimate parameters of Greenspan's mathematical model for spheroids grown in normoxia and hypoxia.** (a-e) Posterior densities for Greenspan model parameters: (a) $R_o(0)$, (b) $R_c$, (c) $\mathcal{R}$, (d) $s$, (e) $\lambda$. Prediction intervals for (f) normoxia. (g) hypoxia. In (f-g) colour bands, in decreasing darkness, represent 50%, 75%, 95%, 97.5%, and 99.5% prediction intervals.

**Fig Y: Additional results from deoxygenation experiments for WM983b spheroids.** Measurements of (a) $\xi_n(t) = R_n(t)/R_o(t)$, (b) $\xi_i(t) = R_i(t)/R_o(t)$, and (c) $\xi_p(t) = R_p(t)/R_o(t)$.

**Fig Z: Prediction intervals for (a) deoxygenation experimental data and (b) deoxygenation experimental data with additional synthetic data points at** $t$ = 2.5 [**days**]. In (a-b) colour bands, in decreasing darkness, represent 50%, 75%, 95%, 97.5%, and 99.5% prediction intervals. Additional synthetic data at $t$ = 2.5 not shown.

**Fig AA: Direction of movement of necrotic core to edge of spheroid appears random in WM983b re-oxygenation experiments.** (a) Schematic for re-oxygenation experiment, with $t_s$ = 2.5 [days]. (b) Exemplar experimental brightfield images of a single spheroid with image processing to detect spheroid boundary and centroid (red) and necrotic core boundary and centroid (cyan). (c) Trajectory of centroid of spheroid for spheroid imaged in (b). (d) Trajectory of centroid of necrotic core for spheroid imaged in (b). (e) Five exemplar trajectories of the centroid of the necrotic core relative to the centroid of the spheroid. In (e) the thick blue trajectory corresponds to spheroid imaged in (b).

**Fig BB: Additional results for WM793b spheroids. (a-d) Mechanisms governing tumour spheroid growth in normoxia and hypoxia.** (a) Box chart for estimated outer radius when necrotic region forms, $R_c$ [μm]. (b) Box chart for estimated oxygen consumption rate, $\alpha$ [m$^3$ kg$^{-1}$ s$^{-1}$]. (c) Comparison of measured and predicted $R_p(t)$ when pimonidazole staining is present. Note this does not include images where pimonidazole staining is present but does not surround the necrotic core, for example Day 8 of Fig G. (d) Box chart for estimated oxygen partial pressure defining inhibited region from hypothesis 1, $p_i$ [%]. (e-i) Experimental data and prediction intervals for (e) normoxia, (f) hypoxia, (g) deoxygenation, (h) re-oxygenation with $t_s$ = 2 [days], and (i) re-oxygenation with $t_s$ = 4 [days].

**Fig CC: Additional results for WM164 spheroids. (a-d) Mechanisms governing tumour spheroid growth in normoxia and hypoxia.** (a) Box chart for estimated outer radius when necrotic region forms, $R_c$ [μm]. (b) Box chart for estimated oxygen consumption rate, $\alpha$ [m$^3$ kg$^{-1}$ s$^{-1}$]. (c) Comparison of measured and predicted $R_p(t)$ when pimonidazole staining is present. (d) Box chart for estimated oxygen partial pressure defining inhibited region from hypothesis 1, $p_i$ [%]. (e-i) Experimental data and prediction intervals for (e) normoxia, (f) hypoxia, (g) deoxygenation, (h) re-oxygenation with $t_s$ = 2 [days], and (i) re-oxygenation with $t_s$ = 4 [days].

**Table A: Number of spheroids imaged with confocal microscopy for the WM983b, WM793b, and WM164 cell lines.** For WM983b re-oxygenation experiments, denoted by $^*$, we focus on brightfield images.

**Table B: Greenspan's model parameters and MCMC diagnostics for normoxia experiments.** Summary statistics of the MCMC chains include: mean; standard deviation, $\sigma$; and, 25%, 50%, and 75% quartiles, $Q_{25\%}$, $Q_{50\%}$, and $Q_{75\%}$, respectively. To assess convergence of the MCMC chains we compute the potential scale reduction factor, $\hat{R}$, [56] where convergence corresponds to $\hat{R} < 1.1$.

**Table C: Greenspan's model parameters and MCMC diagnostics for hypoxia experiments.** Summary statistics of the MCMC chains include: mean; standard deviation, $\sigma$; and, 25%, 50%, and 75% quartiles, $Q_{25\%}$, $Q_{50\%}$, and $Q_{75\%}$, respectively. To assess convergence of the MCMC chains we compute the potential scale reduction factor, $\hat{R}$, [56] where convergence corresponds to $\hat{R} < 1.1$.

**Table D: Deoxygenation model parameters and MCMC diagnostics for deoxygenation experiments.** Summary statistics of the MCMC chains include: mean; standard deviation, $\sigma$; and, 25%, 50%, and 75% quartiles, $Q_{25\%}$, $Q_{50\%}$, and $Q_{75\%}$, respectively. To assess convergence of the MCMC chains we compute the potential scale reduction factor, $\hat{R}$, [56] where convergence corresponds to $\hat{R} < 1.1$.

**Table E: Re-oxygenation model parameters and MCMC diagnostics for re-oxygenation experiments with $t_s$ = 2 [days].** Summary statistics of the MCMC chains include: mean; standard deviation, $\sigma$; and, 25%, 50%, and 75% quartiles, $Q_{25\%}$, $Q_{50\%}$, and $Q_{75\%}$, respectively. To assess convergence of the MCMC chains we compute the potential scale reduction factor, $\hat{R}$, [56] where convergence corresponds to $\hat{R} < 1.1$.

**Table F: Re-oxygenation model parameters and MCMC diagnostics for re-oxygenation experiments with $t_s$ = 4 [days].** Summary statistics of the MCMC chains include: mean; standard deviation, $\sigma$; and, 25%, 50%, and 75% quartiles, $Q_{25\%}$, $Q_{50\%}$, and $Q_{75\%}$, respectively. To assess convergence of the MCMC chains we compute the potential scale reduction factor, $\hat{R}$, [56] where convergence corresponds to $\hat{R} < 1.1$.
(PDF)

**S1 Movie. Necrotic core movement and removal in WM983b spheroids in response to re-oxygenation.** Time-lapse brightfield microscopy movie for three days following re-oxygenation at $t_s$ = 2.5 [days]. The necrotic core of the spheroid is initially visible as a dark central region. As time progresses the necrotic core is located closer to the edge of the spheroid and the symmetric internal structure is lost. At later times the necrotic core appears to exit the spheroid as a single object.
(MP4)

**S2 Movie. Necrotic core movement in WM983b spheroids in response to re-oxygenation.** Time-lapse brightfield microscopy movie for seven days following re-oxygenation at $t_s$ = 5.5 [days]. The necrotic core of the spheroid is initially visible as a dark central region. At later times the necrotic core is close to the edge of the spheroid but does not exit as a single object. As the spheroid grows necrotic matter forms at the centre of the spheroid and appears to merge with the necrotic matter located closer to the periphery.
(MP4)

**S3 Movie. Loss of the necrotic core in WM164 re-oxygenation experiments.** Time-lapse brightfield microscopy movie for three days following re-oxygenation at $t_s$ = 2.5 [days]. The necrotic core of the spheroid is initially visible as a dark central region. As time progresses mass from the necrotic core at the centre of the spheroid moves towards the periphery and exits the spheroid. The spheroid then appears to resume growth.
(MP4)

## Acknowledgments

We thank Dr Alexander P. Browning and Dr Patrick B. Thomas for helpful discussions, and John Blake for guidance using IncuCyte. This research was carried out at the Translational

Research Institute (TRI), Woolloongabba, QLD. TRI is supported by a grant from the Australian Government. We thank the staff in the microscopy core facility at TRI for their technical support. We thank Prof. Atsushi Miyawaki, RIKEN, Wako-city, Japan, for providing the FUCCI constructs, Prof. Meenhard Herlyn, The Wistar Institute, Philadelphia, PA, for providing the cell lines.

## Author Contributions

**Conceptualization:** Ryan J. Murphy, Gency Gunasingh, Nikolas K. Haass, Matthew J. Simpson.

**Data curation:** Ryan J. Murphy, Gency Gunasingh.

**Formal analysis:** Ryan J. Murphy.

**Funding acquisition:** Nikolas K. Haass, Matthew J. Simpson.

**Investigation:** Ryan J. Murphy, Gency Gunasingh.

**Methodology:** Ryan J. Murphy, Gency Gunasingh, Nikolas K. Haass, Matthew J. Simpson.

**Project administration:** Nikolas K. Haass, Matthew J. Simpson.

**Resources:** Gency Gunasingh, Nikolas K. Haass, Matthew J. Simpson.

**Software:** Ryan J. Murphy.

**Supervision:** Matthew J. Simpson.

**Validation:** Ryan J. Murphy, Gency Gunasingh.

**Visualization:** Ryan J. Murphy.

**Writing – original draft:** Ryan J. Murphy.

**Writing – review & editing:** Ryan J. Murphy, Gency Gunasingh, Nikolas K. Haass, Matthew J. Simpson.

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
