## [Decision Letter · Decision Letter 0]

12 Sep 2022

Dear Dr Murphy,

Thank you very much for submitting your manuscript "Growth and adaptation mechanisms of tumour spheroids with time-dependent oxygen availability" for consideration at PLOS Computational Biology.

As with all papers reviewed by the journal, your manuscript was reviewed by members of the editorial board and by several independent reviewers. In light of the reviews (below this email), we would like to invite the resubmission of a significantly-revised version that takes into account the reviewers' comments.

We cannot make any decision about publication until we have seen the revised manuscript and your response to the reviewers' comments. Your revised manuscript is also likely to be sent to reviewers for further evaluation.

Sincerely,

Philip K Maini

Academic Editor

PLOS Computational Biology

Feilim Mac Gabhann

Editor-in-Chief

PLOS Computational Biology

Reviewer's Responses to Questions

**Comments to the Authors:**

Reviewer #1: This is a very interesting study about an area of special importance to researchers who utilise spheroids for research. Over-all, I think this is solidly put together and has interesting findings, but I worry slightly about over-interpretation of borderline results, and would like some of these aspects clarified. In no particular order, I have the following suggestions and queries:

1. As a general guiding principle, I am a strong believer that Occam's razor should always be observed when constructing mathematical models and entities should not be multiplied without necessity. The reason spheroid models have been relatively simply to date is because, as the authors observe, we are typically dealing with snapshots and our existing models describe these well in terms of oxygen-limited diffusion. What the authors propose in interesting (and indeed, even likely) but I would caution tempering some of the claims or providing more evidence for them.

2. The authors state "In phase (ii) cells in the central region of the spheroid arrest in G1 phase while cells at the periphery continue to proliferate resulting in inhibited growth (Figure 1b). This arrested region is thought to arise due spatial differences in nutrient availability, possibly oxygen, and/or a build up of metabolic waste from cells. In phase (iii) the spheroid is characterised by three regions: a central region composed of a necrotic core, 0 < r < Rn(t); an intermediate region of living but proliferation-inhibited cells" - this seems somewhat contradictory to what is typically observed in both spheroids and in situ tumours. One of the distinguishing features of tumour cells is their extreme hardiness and resistance to quiescence (10.1016/j.cell.2011.02.013 and many other publications). In terms of oxygen, cancer cells continue replication unimpeded up to extremely low oxygen pressure of around ~0.5mmHg before mitotic arrest (10.1093/jnci/93.4.266), which was also seen with spheroids in references the authors cite (10.1371/journal.pone.0153692) and has also been seen in situ cancers, including glioblastoma (10.1038/s41416-020-1021-5): while a proliferation impeded region does exist for non-necrotic cells in spheroids at oxygen pressures below the necrotic region, it has a very small radial extent. My suspicion is that the authors may not be considering the image analysis of their fluorescence data and fully de-convoluting it to avoid bleed-through; this would involve analysing expression across all channels and subtracting a noise-gated version. MATLAB allows this pretty easily, but the figure in 1c looks like a tumour ellipsoid which has an entirely different oxygen profile than a spheroid (10.1098/rsif.2018.0256) so I'd like the authors to address these aspects if they could.

3. Part of the problem might be the use of pimonidazole staining as a proxy for hypoxia - although it has long been used, there are problems in interpreting it (10.1667/RR1305.1, 10.1016/s0167-8140(03)00010-0) as its binding properties fluctuate greatly with pO2, unlike EF5 which remains constant. Have the authors taken stock of this? If its possible to replicate the experiments, I'd suggest doing so with EF5 because of the problems Koch describes with the staining agent itself; if it varies with pO2, then experiments on hypoxia with it might be badly confounded.

4. The necrotic core "exiting" the spheroid seems rather interesting. The video appears to be bright-field microscopy, and I have an idea of what might be happening: in the original work of Sutherland and Fryer etc, they demonstrated that spheroids eventually split apart. The necrotic cells within the core lyse, and perturbations in the suspending fluid mean unbalanced pressures occur, with the spheroid "bursting". I believe that is what the investigators are seeing, but at this point the clump of cells can not longer be said to be a spheroid. From the methods section, I don't seem to think the spheroids were made in a spinner culture or similar (10.1517/14712598.2012.707181) so it is unlikely they will maintain their approximately spherical state. The non-bursting video seems to show the spheroid becoming an ellipsoid (10.1098/rsif.2018.0256) so the figures showing no morphogenic changes from a spherical shape in figure 1 are misleading; if the shape of the spheroid departs from approximately spherical symmetry due to unbalanced pressures or irregularities in growth media, it will very quickly depart from the assumptions of perfect diffusion. The bubbles on the bright field microscopy indicate some media disturbances too.

5. I do not think, based on the considerations of the above points, that the authors have proven "Oxygen diffusion alone is insufficient to describe spheroid growth" - if anything, all of the phenomena they describe in this work are consistent with pO2 limited diffusion. It would be interesting to find other factors (which presumably must exist) but as it stands I do not think the authors have overcome the threshold of proof to demonstrate this.

6. Greenspan's model, it must be noted, is a very old but useful general model for spherical oxygen diffusion, but it has long since been superseded. However, following on from point 4, the primary assumption that "The model assumes each spheroid is spherically symmetric and maintained by cell-cell adhesion or surface tension" is quite clearly contradicted by the evidence the authors have provided; in this case, all spherically symmetric calculations rapidly break down, but that would be predicted by the nature of oxygen diffusion itself and basically amounts to a solution of poisson's equation is a unique geometry.

7. As I understand it, the results shown in figure 2 are consistent for the projected and measured growth dynamics in multiple cell lines shown in (doi.org/10.1371/journal.pone.0153692) - using this model or similar, you would expect these growth dynamics, but the lack of radial measures in the figure makes it difficult to directly compare and should be labelled. The general growth dynamics of spheroids are well understood including their phases (Cancer Res 1983, 43(2): 556–560, doi not readily available) and in all cases, a sigmoid function is typically obtained (10.1111/j.1365-2184.1994.tb01407.x) - however, from what I can observe, none of the spheroids were grown to plateau stage, which unfortunately reduces ones ability to make strong generalisations from the presented data. Spheroids in this work seem very young and small, and it is difficult to see differences in properties until they grow considerably, as indeed their quasi-linear / plateau point varies by OCR and pO2 at spheroid edge.

8. Figures 3 and 4 and very difficult to follow, and the caption barely helps. If you are cross comparing two hypotheses, it is might be easier to include a table in your graphic explicitly stated differences.

9. Again, the adaptations to hypoxia is in my opinion more readily explained by simple non-sphericity than by esoteric mechanisms. I have no doubt that processes like waste disposal play some role, but on the basis of the evidence presented, I do not feel the authors have even considered the huge role a departure from sphericity has on re-oxygenation and even the physical stability of spheroids.

While I commend the authors for their work so far, I think these objections would have to be either addressed or conceded before this work can be published. Occam's razor should not be forgotten, and explanations requiring the least number of additional assumptions should be favoured, at least unless there is solid evidence that this is not sustainable. In this case, I am not yet convinced by the evidence presented. Oxygen diffusion historically explains the dynamics of avascular tumours and spheroids to the levels we have been able to measure with probes and stains. Due to experimental limitations, this is generally the level to which we can resolve. The authors assertions that diffusion is insufficient to explain what they observed I believe is less likely than the competing idea that their model is overly reliant on spherical symmetry compounded with difficultly interpreting confocal staining. I am of course willing to see revisions and again congratulate the authors on an ambitious undertaking.

Reviewer #2: no further comments

Reviewer #3: Murphy et al use an extended Greenspan model for necrotic tumour development to investigate how temporal variation in conditions influences the development of a tumour. Specifically they add temporal variability to existing model parameters (for deoxygenation and re oxygenation) and then fit the model (under various hypotheses) to experimental data where the oxygenation of the tumour spheroid is varied over time.

The paper is well written and well presented. The topic is interesting to a general audience and, while I can’t vouch for the experimental component, the modelling is sound and the parameter fitting is appropriate and well explained.

I would have liked to see more details on the parameter fits in the main paper but understand their position in the SI.

There a few minor issues I have identified below.

Figure 2 I think it should be a) and d) which show snapshots

Figure 2 What are the blue and orange lines in b and e is one a fit? Its not clear at preasent

**Have the authors made all data and (if applicable) computational code underlying the findings in their manuscript fully available?**

Reviewer #1: Yes

Reviewer #2: Yes

Reviewer #3: Yes

PLOS authors have the option to publish the peer review history of their article (what does this mean?). If published, this will include your full peer review and any attached files.

Reviewer #1: **Yes: **Anonymous

Reviewer #2: No

Reviewer #3: **Yes: **James Osborne
---

## [Decision Letter · Decision Letter 1]

21 Dec 2022

Dear Dr Murphy,

We are pleased to inform you that your manuscript 'Growth and adaptation mechanisms of tumour spheroids with time-dependent oxygen availability' has been provisionally accepted for publication in PLOS Computational Biology.

Best regards,

Philip K Maini

Academic Editor

PLOS Computational Biology

Feilim Mac Gabhann

Editor-in-Chief

PLOS Computational Biology

Reviewer's Responses to Questions

**Comments to the Authors:**

Reviewer #1: I think the caveats improve the paper immensely - happy to accept now as is.

**Have the authors made all data and (if applicable) computational code underlying the findings in their manuscript fully available?**

Reviewer #1: Yes

PLOS authors have the option to publish the peer review history of their article (what does this mean?). If published, this will include your full peer review and any attached files.

Reviewer #1: **Yes: **David Robert Grimes

---

## [Editor Report · Acceptance letter]

4 Jan 2023

PCOMPBIOL-D-22-00961R1 

Growth and adaptation mechanisms of tumour spheroids with time-dependent oxygen availability

Dear Dr Murphy,

I am pleased to inform you that your manuscript has been formally accepted for publication in PLOS Computational Biology. Your manuscript is now with our production department and you will be notified of the publication date in due course.

With kind regards,

Olena Szabo
